# Event boundaries shape temporal organization of memory by resetting temporal context

Yi Pu [1✉], Xiang-Zhen Kong[2✉], Charan Ranganath [3,4] & Lucia Melloni [1,5✉]

In memory, our continuous experiences are broken up into discrete events. Boundaries between events are known to influence the temporal organization of memory. However, how and through which mechanism event boundaries shape temporal order memory (TOM) remains unknown. Across four experiments, we show that event boundaries exert a dual role: improving TOM for items within an event and impairing TOM for items across events. Decreasing event length in a list enhances TOM, but only for items at earlier local event positions, an effect we term the local primacy effect. A computational model, in which items are associated to a temporal context signal that drifts over time but resets at boundaries captures all behavioural results. Our findings provide a unified algorithmic mechanism for understanding how and why event boundaries affect TOM, reconciling a long-standing paradox of why both contextual similarity and dissimilarity promote TOM.

[1] Department of Neuroscience, Max Planck Institute for Empirical Aesthetics, Frankfurt am Main, Germany. [2] Department of Psychology and Behavioral Sciences, Zhejiang University, Hangzhou, China. [3] UC Davis Center for Neuroscience, University of California, Davis, CA, USA. [4] Department of Psychology, University of California, Davis, CA, USA. [5] Department of Neurology, NYU Grossman School of Medicine, New York City, NY, USA. ✉email: yi.pu@ae.mpg.de; xiangzhen.kong@zju.edu.cn; lucia.melloni@ae.mpg.de

Episodic memory is temporally organized[1]. In memory, it is easier to differentiate between items that are far than items that are close in time (i.e., the temporal distance effect, e.g.,[2]). Substantial evidence suggests that the temporal organization of episodic memory can be influenced by various types of salient and meaningful changes in ex/internal environments (i.e., event boundaries)[3,4]. However, it remains unknown how and why event boundaries affect temporal order memory (TOM).

Recent work has shown that TOM is more accurate for items presented within an event than for items that are separated by an event boundary, even when controlling for the time elapsed between items (e.g.,[5–7]). Yet, how event boundaries shape TOM remains unknown. According to the Event Horizon Model[8–10], segmenting a continuous experience, such as a fixed-length list into smaller events separated by event boundaries should result in less competition among items in an event, but also more competitions among different events. However, it is unclear how the effects on interference could impact TOM. For instance, it remains to be examined whether boundaries enhance and/or impair TOM for items within and across events, respectively; and how event length (or the number of event boundaries) in a fixed-length list shapes TOM for within and across-event items.

One defining feature of episodic memory is that when encoding an experience, individuals link the experience with its surrounding spatiotemporal context (e.g.,[11]). As a consequence, the encoded spatiotemporal context triggers recollection of the specific experience and vice versa. Building on this observation, a class of computational models (i.e., temporal context models) posit that items are associated together through a gradually changing temporal context, such that the contextual representations are more dissimilar for items far away than for items close in time[12–14]. This class of models have then been used to explain the temporal distance effect of TOM, i.e., better TOM for a longer lag than for a shorter lag (e.g.,[2,15]), by assuming that contextual dissimilarity facilitates memory differentiation. Consistent with this account, neuroscientific research shows a gradient of similarity in hippocampal representations across long time periods, such that hippocampal representations are more dissimilar for experiences further apart than experiences close in time (e.g.,[16–18]). Thus, hippocampal representations are thought to serve as a neural correlate of temporal context that binds events together[1].

Recently, studies also show that our mental and neural representations can change abruptly at event boundaries[19,20], leading to more similar mental and neural representations for within-event items than for across-event items (e.g.,[19]). Therefore, the boundary effect of better within-event TOM than across-event TOM is explained by assuming that contextual similarity promotes memory[5,21,22]. Thus, at the computational level, the temporal distance effect and the boundary effect pose a paradox: why is it that sometimes contextual dissimilarity benefits TOM and sometimes contextual similarity benefits TOM[23]? Different theoretical and computational models have been put forward to explain either of the two effects, but none of them, to our knowledge, can explain both. Therefore, there is a need to reconcile the disparate accounts and paradoxical findings at the algorithmic level to account for how spatiotemporal context influences TOM.

Here we report four experiments aimed at understanding how event boundaries and temporal distance affect the temporal organization of memory, and introduce a computational model to explain behavioural results. In these experiments, we presented participants with sequences of objects, whose order they were asked to remember. We subsequently tested their temporal order memory. We manipulated the presence and the number of event boundaries (i.e., event length) in a fixed-length list, while controlling for the absolute list position. Event boundaries were operationalized as perceptual/context shifts (i.e., colour change of the background frame[22]).

We find that event boundaries exert a dual effect: they simultaneously enhance TOM for items within events and impair TOM for items across events. We also find that decreasing event length by increasing the number of event boundaries in a fixed-length list enhances TOM. Yet, this effect is limited to items occupying earlier local positions for both within and across-events. We term this phenomenon the local primacy effect. Finally, we find that the boundary effect and the temporal distance effect, two effects thought to be computationally incompatible, can co-exist in the same experiment, and moreover, that event boundaries affect TOM above and beyond a mere temporal distance effect.

To account for these results, we have developed a computational model based on an existing class of temporal context models[12,13,24], in which boundaries reset the temporal context signal by reinstating a certain proportion of the first contextual representation. Our model explains both the boundary effect and the temporal distance effect, reconciling the two paradoxical effects in memory literature, while also accounting for the local primacy effect and all the other findings in the present study. Our model provides two key insights: first, event boundaries do not cause a random sharp change in contextual representations as previously hypothesized[5,23]. Instead, event boundaries cause a systematic change in temporal context by recovering a certain proportion of the first contextual representation. Second, the latent cause for better TOM is not increased contextual similarity or dissimilarity of the two probed items, as assumed by previous theoretical and computational models[5,21,22]; instead, it is driven by an increased difference between the contextual representations of the probed items relative to that of the first item in the list. Overall, the present study reveals important behavioural phenomena and provides a unified account of how and why organizational processes during encoding structure the temporal organization of memory.

## Results

**Event boundaries affect TOM by playing a dual role**. Our first experiment was aimed at examining how event boundaries shape TOM: do they enhance and/or impair TOM for within and across-event items? To address this question, we presented participants with sequences of objects whose order they had to memorize. The design for Experiment 1, built on previous studies (e.g.,[6,7]), is schematically depicted in Fig. 1a. We exposed participants to two sequence types: one in which the coloured frame changed every six images (i.e., boundary condition), and the other in which the coloured frame remained constant across all 36 images (i.e., no boundary condition). In the subsequent testing phase, participants made recency judgments on pairs of objects chosen from the just-encoded sequence. For the boundary condition, the probed object pairs were either from the same event (i.e., same background colours, called within-event pairs) or from two adjacent events (i.e., different background colours, called across-event pairs, Fig. 1b). Probed pairs in the no boundary condition matched list positions to those in the boundary condition. The experimental procedure in Experiment 1 was identical for the rest of experiments, unless otherwise specified.

We had four trial types—in the boundary condition, pairs of objects were either in the same event or separated by an event boundary, and in the no boundary baseline condition, pairs of objects were tested so as to match the same list positions tested in the boundary condition. The no boundary condition served as a baseline condition, which enabled us to explicitly investigate the

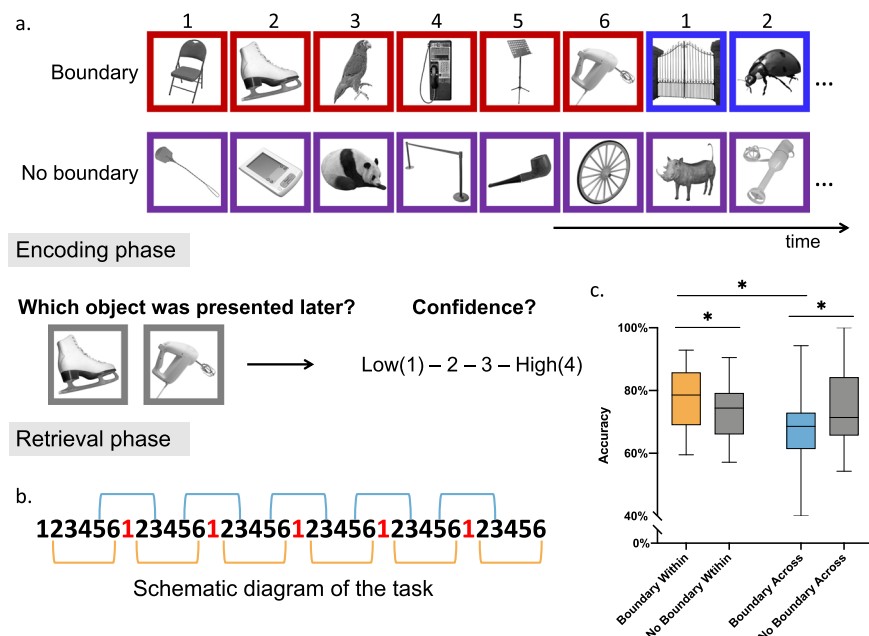

**Fig. 1 Experimental paradigm. a** Experimental task. 36 grey-scaled trial-unique object images embedded in a coloured frame were sequentially presented to participants. In the boundary condition, the colour of the frame was consistent for six consecutive images before switching to another one, while in the no boundary condition, the colour of the frame was consistent for all 36 objects. Each object image was presented to participants for 2.5 s, preceded by 0.5 s fixation cross and followed by 2 s inter-trial interval, during which the coloured frame stayed on the screen. In the boundary condition, event boundaries were defined as the trial in which the colour of the frame updated with the co-occurring object. Immediately after encoding the 36 images, participants made recency judgments on pairs of items from the just-encoded sequence. Each recency judgment was then followed by a confidence rating for each decision on a four-point scale. Participants had a self-paced short break after finishing each sequence. The images used were taken from publicly available dataset (Bank of Standardized stimuli, BOSS, https://sites.google.com/site/bosstimuli/, © 2010 Brodeur et al. & © 2014 Brodeur et al.)[49,50]. **b** Schematic diagram of the task in Experiment 1. Numbers in red depict event boundaries. There were two pair types marked by yellow and blue square brackets, denoting within-event pairs and across-event pairs respectively (short for Boundary Within and Boundary Across in the figure respectively). The two pair types were separated by the same number of intervening items. In the no boundary condition, item pairs took identical list positions as in the boundary condition (short for No Boundary Within and No Boundary Across respectively). **c** Group averaged temporal order memory for within and across-event pairs in the boundary vs. no boundary conditions in Experiment 1 ($n = 26$). A repeated measures ANOVA on the accuracy of recency judgments showed a significant interaction between Condition and List Position (F (1, 25) = 18.05, $p < 0.001$, $\eta^2 = 6.364\%$), as well as a main effect of List Position (F(1,25) = 5.339, $p = 0.0294$, $\eta^2 = 3.452\%$). Simple effect analyses showed that TOM was significantly better for within than across-event pairs in the boundary condition (t(25) = 5.216, $p < 0.001$, two-sided, $q < 0.05$, FDR corrected for multiple comparisons), and was significantly better for within-event pairs (t(25) = 2.418, $p = 0.0232$, two-sided, $q < 0.05$, FDR corrected) and significantly worse for across-event pairs (t(25) = −4.022, $p < 0.001$, two-sided $q < 0.05$, FDR corrected) in the boundary condition compared to matched pairs in the no boundary condition. The boxes in box plots show the inter-quartile range (IQR) and the median. Whiskers in box plots represent the minimum and maximum in the dataset. The asterisk (*) represents statistical significance at p < 0.05. Source data are provided as a Source Data file.

effect of boundaries on TOM for within and across-events pairs. We reasoned that if event boundaries influence TOM, we would expect an interaction between Condition (boundary vs. no boundary condition) and List Position ([matched] within-event position vs. [matched] across-event position). Throughout the experiments in the paper, our analyses focused on the accuracy of recency judgments. In Supplementary materials, we reported results on reaction time (Supplementary Fig. 1) and confidence ratings (Supplementary Fig. 2) of recency judgments.

In line with our hypothesis, a repeated measures ANOVA on the accuracy of recency judgments revealed a significant interaction between Condition and List Position (F (1, 25) = 18.05, $p < 0.001$, $\eta^2 = 6.364\%$, Fig. 1c), as well as a main effect of List Position (F(1,25) = 5.339, $p = 0.0294$, $\eta^2 = 3.452\%$). No other significant effect was observed. In follow-up planned contrasts (FDR corrected for multiple comparisons), we first examined whether we could replicate previous findings that TOM was significantly better for within than across-event pairs in the boundary condition ([5,25]). In line with previous findings, we found that TOM was significantly better for within than across-event pairs in the boundary condition (t(25) = 5.216, $p < 0.001$, two-sided, $q < 0.05$, 95% Confidence

Interval (CI) = 6.208 to 14.31). We noted that the absolute list positions were not perfectly matched between within and across event pairs. In particular, within-event pairs took both first and last positions in the long list. As such, the primacy and recency positions might have memory advantages in the long list (e.g., primacy and recency effect[26]), and directly comparing the within vs. across-event TOM might bias the results in favour of within-event TOM. To rule out this potential confound, we excluded the first and last pairs from within-event pairs and re-ran the previous analysis. The result remained significant (t(25) = 3.022, $p = 0.0057$, two-sided, 95% CI = 2.702 to 14.26), indicating that the boundary effect is robust to list position, and does not merely reflect the primacy and recency effect in a long list. Then, we compared TOM in the matched positions in the no-boundary condition. However, no differential effect was present (t(25) = −1.009, $p = 0.3225$, two-sided, 95% CI = −5.607 to 2.494), suggesting that event boundaries do play a role in affecting TOM. Next, we compared TOM in the boundary condition against TOM in the no boundary condition, controlling for list position. We found that TOM was significantly better for within-event pairs (t(25) = 2.418, $p = 0.0232$, two-sided, $q < 0.05$, 95% CI = 0.7042 to 8.805) and significantly worse for

across-event pairs (t(25) = −4.022, $p < 0.001$, two-sided $q < 0.05$, 95% CI = −9.554 to -1.453) in the boundary condition compared to matched pairs in the no boundary condition. These results demonstrate that event boundaries exert a dual influence on TOM, simultaneously enhancing TOM for within-event pairs and impairing TOM for across-event pairs.

**Decreasing event length also yields a robust boundary effect, which can co-exist with the temporal distance effect in the same experiment**. Having established a dual role of event boundaries in affecting TOM, we next examined (1) whether decreasing event length by increasing the number of event boundaries in a list could also generate a robust boundary effect; and (2) whether the boundary effect and the temporal distance effect, two effects thought to be computationally incompatible could co-exist in the same experiment. A schematic diagram of the task is depicted in Fig. 2a. In this experiment, the colour of the frame changed after every four images. Thus, each event was shorter and there were more event boundaries in this experiment compared to Experiment 1.

To investigate the boundary effect and the temporal distance effect, we probed within and across-event pairs separated by one intervening item (lag1), as well as across-event pairs separated by three intervening items (lag3). A one-way repeated measures ANOVA on the accuracy of recency judgments revealed a significant effect of Pair Type (F(1.448, 37.64) = 18.63, $p < 0.001$, $R^2 = 25.47\%$) (Fig. 2b). To determine the robustness of the boundary effect to event size, we compared TOM for within against across-event pairs at lag1. TOM was significantly better for within-event pairs than across-event pairs (t(26) = 4.820, $p < 0.001$, two-sided, q < 0.05, 95% CI = 12.87 to 31.99). The result should not be confounded with list position, since a significant effect of boundary still held, even when excluding the first and the last two pairs from the within-event pairs (t(26) = 4.283, $p < 0.001$, two-sided, 95% CI = 10.92 to 31.07). These results above demonstrate a robust boundary effect with events of a smaller size.

We next examined the temporal distance effect, comparing TOM for across-event pairs at lag3 against lag1. TOM was significantly better for across-event pairs at lag3 than for across-event pairs at lag1 (t(26) = 2.941, $p < 0.001$, two-sided, q < 0.05, 95% CI = 2.987 to 16.84), replicating the well-established temporal distance effect. To explore whether event boundaries affect TOM beyond the temporal distance, we compared TOM for within-event pairs at lag1 against across-event pairs at lag3. TOM was significantly better for within-event pairs at lag1 than for across-event pairs at lag3 (t(26) = 4.524, $p < 0.001$, two-sided, q < 0.05, 95% CI = 6.825 to 18.19). The significant result held when the first trial and the last two trials were excluded from the within-event pairs (t(26) = 3.784, $p < 0.001$, two-sided, 95% CI = 4.997 to 16.88). These results indicate that event boundaries structure the temporal organization of memory above and beyond a mere temporal distance effect; and provide the important insight that the boundary effect and the temporal distance effect, two effects thought to be computationally incompatible, can co-exist in the same experiment, aligning with previous studies showing the co-existence of the two effects (e.g.,[25]).

**Decreasing event length enhances TOM due to the local primacy effect caused by event boundaries**. Having observed a robust boundary effect even when event length decreases, in Experiment 3, we further investigated whether and how event length affects TOM performance. When holding the number of list items constant, decreasing the length of each event necessarily increases the number of events (and event boundaries) in a list.

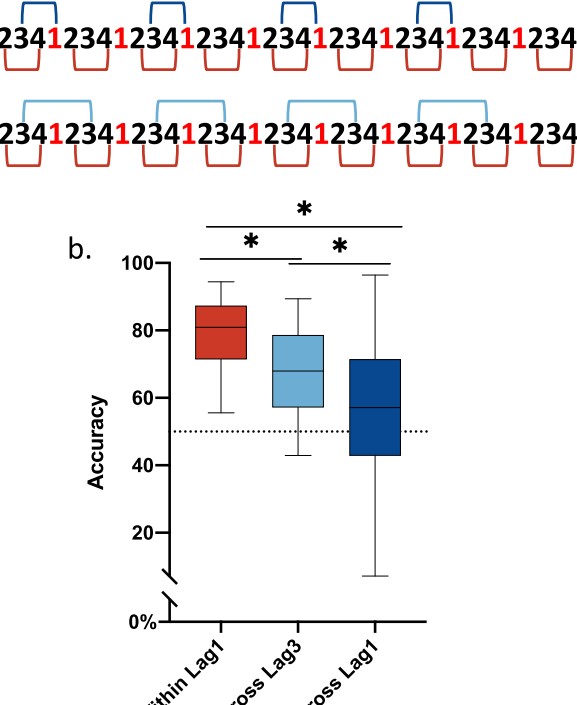

**Fig. 2 Design and results of Experiment 2. a** Schematic diagram of the task in Experiment 2. The sequence of 36 images was segmented as 4 items per event. Three pair types were tested for recency judgments, marked by red, light blue, and blue square brackets, representing within-event pairs with one intervening item (short for Within Lag1 in the figure), across-event pairs with three intervening items (short for Across Lag3 in the figure) and across-event pairs with one intervening item (short for Across Lag1 in the figure), respectively. **b** Box plots of group averaged temporal order memory for within and across-event pairs for lag1 and lag3 ($n = 27$). A one-way repeated measures ANOVA on the accuracy of recency judgments showed a significant effect of Pair Type (F(1.448, 37.64) = 18.63, $p < 0.001$, $\eta^2 = 25.47\%$). Simple effect analyses showed that TOM was significantly better for within-event pairs than across-event pairs (t(26) = 4.820, $p < 0.001$, two-sided, q < 0.05, FDR corrected for multiple comparisons), for within-event pairs at lag1 than for across-event pairs at lag3 (t(26) = 4.524, $p < 0.001$, two-sided, q < 0.05, FDR corrected), and for within-event pairs at lag1 than for across-event pairs at lag3 (t(26) = 4.524, $p < 0.001$, two-sided, q < 0.05, FDR corrected). Numbers in red denote event boundaries. The boxes show the inter-quartile range (IQR) and the median. Whiskers in box plots represent the minimum and maximum in the dataset. The asterisk (*) represents statistical significance at $p < 0.05$. Source data are provided as a Source Data file.

Thus, having shorter events might have the beneficial effect of decreasing the interference among items within each event. However, it may also negatively impact TOM by increasing the competition among different events during retrieval.

To address this question, we created two sequences of fixed length: one containing four items per event (Event 4) and another containing six items per event (Event 6). The Event 4 condition contained more event boundaries than the Event 6 condition (Fig. 3a). To allow for a fair comparison across the two sequences, we created three types of probed pairs controlling for the absolute list positions. The different pair types are marked by the colour-coded square brackets in Fig. 3a. Pair type 1 were within-event pairs, which took both identical list positions and local event positions (2–4) in the two sequences, yet differed by the amount of event boundaries in the long list. Pair type 2 were also within-

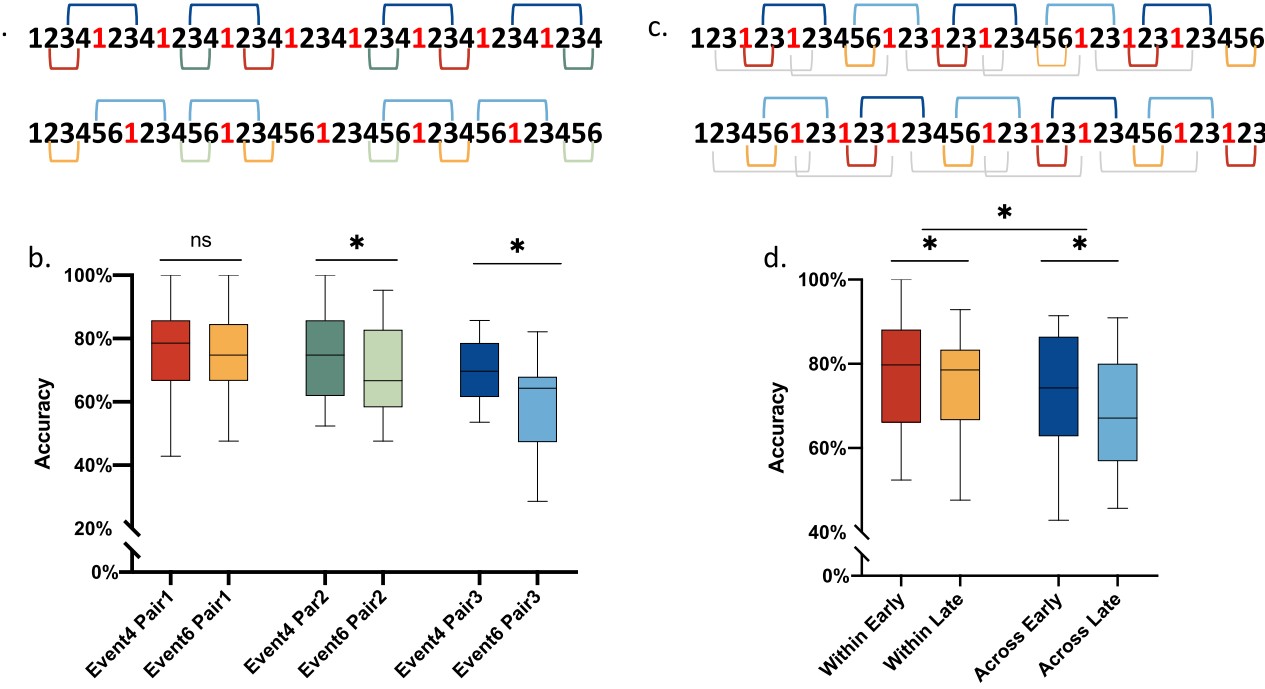

**Fig. 3 Design and results of Experiment 3 & 4. a** Schematic diagram of the task in Experiment 3. Two conditions were tested: one with events containing four items (Event 4) and the other containing six items (Event 6). Three matched pair types were tested: Pair type 1 marked by the red and yellow square brackets, Pair type 2 marked by green and light green square brackets and Pair type 3, marked by blue and light blue square brackets in the two conditions. **b** Box plots of group averaged temporal order memory for the three pair types ($n = 32$). No significant difference was found between the Event 4 condition and the Event 6 condition for pair type 1 ($t(31) = 0.06437$, $p = 0.9491$, two-sided, $q = 0.3322$, FDR corrected for multiple comparisons); however, TOM was significantly better for the Event 4 than for Event 6 condition for both pair type 2 ($t(31) = 2.334$, $p = 0.0262$, two-sided, $q < 0.05$, FDR corrected) and pair type 3 ($t(31) = 3.93$, $p < 0.001$, two-sided, $q < 0.05$, FDR corrected). **c** Schematic diagram of the task in Experiment 4. Two sequence types were tested containing the same number of event boundaries but segmented differently, e.g., 336336... vs. 633633... items per event. Three pair types were tested: within-event pairs taking earlier and later local positions (matched by absolute list positions) denoted by the red and yellow square brackets (short for Within Early and Within Late respectively). Across-event pairs taking earlier and later local positions (matched by absolute list positions) denoted by the blue and light blue square brackets (short for Across Early and Across Late respectively). Across-event pairs with a longer lag than pairs marked by blue and light blue square brackets, are marked by light grey square brackets. d. Box plots of group averaged temporal order memory for within and across-event pairs taking earlier and later positions shown in Fig. 3c ($n = 30$). TOM was significantly better for within vs. across-event pairs ($F(1,29) = 8.217$, $p = 0.0076$, $\eta^2 = 4.657\%$), and for early vs. late pairs ($F(1,29) = 8.933$, $p = 0.0057$, $\eta^2 = 2.503\%$). Planned contrasts further confirmed that TOM was significantly better for earlier vs. later pairs for both pair types (for within event pairs: $t(29) = 2.256$, $p = 0.0318$, two-sided, $q < 0.05$, FDR corrected; for across-event pairs: $t(29) = 3.839$, $p < 0.001$, two-sided, $q < 0.05$, FDR corrected). The boxes show the inter-quartile range (IQR) and the median. Whiskers in box plots represent the minimum and maximum in the dataset. The asterisk (*) represents statistical significance at p < 0.05. Numbers in red denote event boundaries in Fig. 3a, c. Source data are provided as a Source Data file.

event pairs, which took identical list positions, but earlier local positions (2–4) for the Event 4 condition and later local positions (4–6) for the Event 6 condition. Pair type 3 were across-event pairs, matched by the average list positions in the two sequences. Because of different segmentation of the two sequences, the local positions were always earlier in the Event 4 condition compared to the Event 6 condition (i.e., 3'-3 in Event4 vs. 5'-3 in Event6).

A repeated measures ANOVA on the accuracy of recency judgments revealed a significant interaction between Condition and Pair Type (F (1.967, 60.96) = 5.079, $p = 0.0094$, $\eta^2 = 2.084\%$) (Fig. 3b) and a main effect of Condition (F(1,31) = 11.72, $p = 0.0018$, $\eta^2 = 2.823\%$) and Pair Type (F (1.411, 43.73) = 15.24, $p < 0.0001$, $\eta^2 = 10.37\%$). A follow-up planned contrast (FDR corrected for multiple comparisons) revealed no significant difference between the Event 4 condition and the Event 6 condition for pair type 1 (t(31) = 0.06437, $p = 0.9491$, two-sided, q = 0.3322, 95% CI = −4.842 to 4.545). That is, TOM was comparable when sequences were matched both for list positions and local serial positions within an event. However, if local serial positions in events differed across conditions (i.e., for pair types 2 and 3), TOM was

significantly better for the Event 4 than for Event 6 condition for both pair type 2 (t(31) = 2.334, $p = 0.0262$, two-sided, q < 0.05, 95% CI = 0.5638 to 8.362) and pair type 3 (t(31) = 3.93, $p < 0.001$, two-sided, q < 0.05, 95% CI = 4.632 to 14.63).

As shown in Fig. 3b, event length affected TOM only when the pairs consisted of items at earlier serial positions in the Event 4 condition, as compared to those at later serial positions in the Event 6 condition. This pattern of results has not been predicted based on the Event Horizon Model[10]. Based on our results, we put forward a new hypothesis that event boundaries cause an advantage in TOM for earlier local event positions, such that TOM is better for items taking earlier vs. later local event positions, controlling for absolute list positions. Since the absolute list positions in Experiment 3 could not be perfectly matched for across-event pairs in the Event 4 condition and in the Event 6 condition, we run a follow up experiment to directly test this hypothesis while addressing this confound.

A schematic diagram of Experiment 4 is shown in Fig. 3c. We created two types of sequences that contained an identical number of event boundaries but differed in the sequences by

which the events were formed. I.e., in one sequence, event sequences took the form of 3-3-6-3-3-6-3-3-6; while in the other sequence, it was 6-3-3-6-3-3-6-3-3. Thereby, the coloured frame remained constant for either three or six objects before changing to a different one. This experimental manipulation allowed us to create pairs taking earlier vs. later local event positions in the two sequences, while keeping their absolute list positions and the total number of event boundaries identical (Fig. 3c).

In line with our hypothesis, a repeated measures ANOVA on the accuracy of recency judgments with within-subjects factors of Pair Type (within vs. across-event pairs) and Position (earlier vs. later local positions) revealed a significant main effect of Pair Type ($F(1,29) = 8.217$, $p = 0.0076$, $\eta^2 = 4.657\%$) and Position ($F(1,29) = 8.933$, $p = 0.0057$, $\eta^2 = 2.503\%$), with TOM being significantly better for within than across-event pairs as well as for earlier than later pairs (Fig. 3d). No significant interaction was found. Planned contrasts further confirmed that TOM was significantly better for earlier vs. later pairs for both pair types ($t(29) = 2.256$, $p = 0.0318$, two-sided, q < 0.05, 95% CI = 0.2964 to 6.053 for within-event pairs, $t(29) = 3.839$, $p < 0.001$, two-sided, q < 0.05, 95% CI = 2.384 to 8.141 for across-event pairs).

To replicate the temporal distance effect, we compared TOM for across-event pairs separated by more intervening items (Fig. 3c, marked by grey square brackets) vs. across-event pairs separated by fewer intervening items, collapsing over earlier and later positions (i.e., those across-event pairs used in our main hypothesis testing above). As expected, pairs with a longer lag were better remembered than those with a shorter lag ($t(29) = 3.671$, $p < 0.001$, one-tailed, 95% CI = 1.941 to 6.826. One-tailed test was used, since we had a strong a priori about the directionality of this comparison). Of note, list positions cannot be perfectly matched for this contrast, and thus a residual uncertainty as to the effect of list position for this comparison remains.

Altogether, Experiment 3 and 4 uncovered a behavioural phenomenon, which we termed the local primacy effect. This phenomenon explains why varying event length affects TOM.

**Computational modelling.** Our behavioural experiments demonstrated three basic findings: (1) Boundaries play a dual role in shaping TOM by both enhancing within-event TOM and impairing across-event TOM, such that TOM is better for pairs of items within the same event than for pairs in separate events (the boundary effect); (2) TOM is better for items that are farther apart in time, than for items close together in time (the temporal distance effect), which could co-exist with the boundary effect in the same experiment, although the two effects were thought to be computationally incompatible; (3) TOM is better for pairs of items that occur closer to the beginning of an event than for pairs of items that occur later in the event, when the absolute list positions are kept identical (the local primacy effect). These results, however, pose a challenge to mechanistic accounts of TOM. The previous models[5,21] have been put forward to account for the boundary effect, yet none, to our knowledge, aim to simultaneously account for both the boundary effect and the temporal distance, nor for the local primacy effect.

To address this challenge, we developed an algorithmic model aimed at capturing the three effects described above. Our model was built on a class of temporal context models[12,13,24,27], in which items are associated with each other via a context signal. The context signal changes gradually from moment to moment (i.e., slow drift), reflecting subtle changes in the environment or in the subjects' mental state[12]. Since recent neuroimaging and psychophysics studies have shown that event boundaries can cause a sharp change (i.e., shift) in brain/mental state (e.g.,[19,20]), it has been hypothesised that such a sharp shift is caused by a faster random change in temporal

context at event boundary (e.g.,[23]). Our model, however, assumes that the shift in temporal context is not a random process, but instead is based on reinstatement of the pre-experimental context.

This assumption fits with existing theories of event cognition[9], which postulate that following an event boundary participants generate a new event model. Forming an event model is not simply random, instead it is an attempt to make sense of the information that is being processed[28,29]. In complex, real life events, building a new event model often relies on memory retrieval[28–31]. For instance, when crossing a doorway en route to a particular location, it makes sense to reinstate information from the beginning of the journey, in order to recall what made us leave the room in the first place. If mental context randomly shifted at each event boundary, one would possibly forget one's destination after crossing the first doorway. In our and other temporal order memory experiments, events are much simpler, but it is nonetheless sensible to reinstate information from the beginning of the experiment, when participants first learned about the structure of the task (i.e., changing colour frames, sequential structure, etc.), as well as any other schematic information which may facilitate learning of the sequence.

In our model, we operationalized such a systematic change at event boundaries by assuming that a certain proportion of the initial contextual information is reinstated at each event boundary and that the initial contextual information contains both pre-experimental contextual representation (e.g., the task structure) and the first contextual representation of the list. In a later analysis, we compared our model with a model in which a random sharp change occurs at event boundary[5], in the absence of any reinstatement of past contextual states.

Following Estes[13], we operationalized context as a set of binary elements with 100 features (e.g.,[5]). At non-boundary time points, context fluctuates gradually, with active elements (1) turning off and inactive elements (0) turning on with probability $p$, such that

$$C_t = (1 - p)C_{t-1} + pC^{IN} \tag{1}$$

where $C_t$ represents contextual representation at time point t, $\mathbf{C^{IN}}$ represents random noise. When a boundary occurs, the context signal resets at rate $\lambda$ (Fig. 4a), such that

$$C_t = (1 - \lambda)((1 - p)C_{t-1} + pC^{IN}) + \lambda C_1 \tag{2}$$

meaning that a proportion ($\lambda$) of the first contextual representation is reinstated at event boundaries. As shown in Fig. 4b, c, contextual representations, as operationalized in our model, are clustered, such that contextual representations are more similar for pairs of items in the same event than for pairs of items in different events, and they are more similar for temporally proximal items than for temporally distant items.

To operationalize the recency judgment process, we first assumed that participants retrieve the learning contexts associated with each of the probed items[32]. Next, we assumed that the outcome of the recency judgment is based on the sign of the subtraction between the distances of the two retrieved contexts to the first context which serves as a common reference point (i.e., $d2$-$d1$, Fig. 4d). Accuracy would be expected to increase as a function of the difference between $d2$ and $d1$ (see Methods).

Simulations showed that the model outputs mirrored the pattern of the group averaged behavioural results in each experiment (Fig. 5a–d, with a drift rate $p = 0.02$, and a reset rate $\lambda = 0.2$; 1000 different simulation iterations; see Methods for details). Statistically, the model outputs significantly explained the behavioural data across experiments ($R^2 = 0.64$, $p < 0.001$, a generalized linear model (GLM); Fig. 5e). The significance also held ($R^2 = 0.88$, $p < 0.001$) when fitting a generalized linear mixed model (GLMM), in which experiments/sequences were set as a

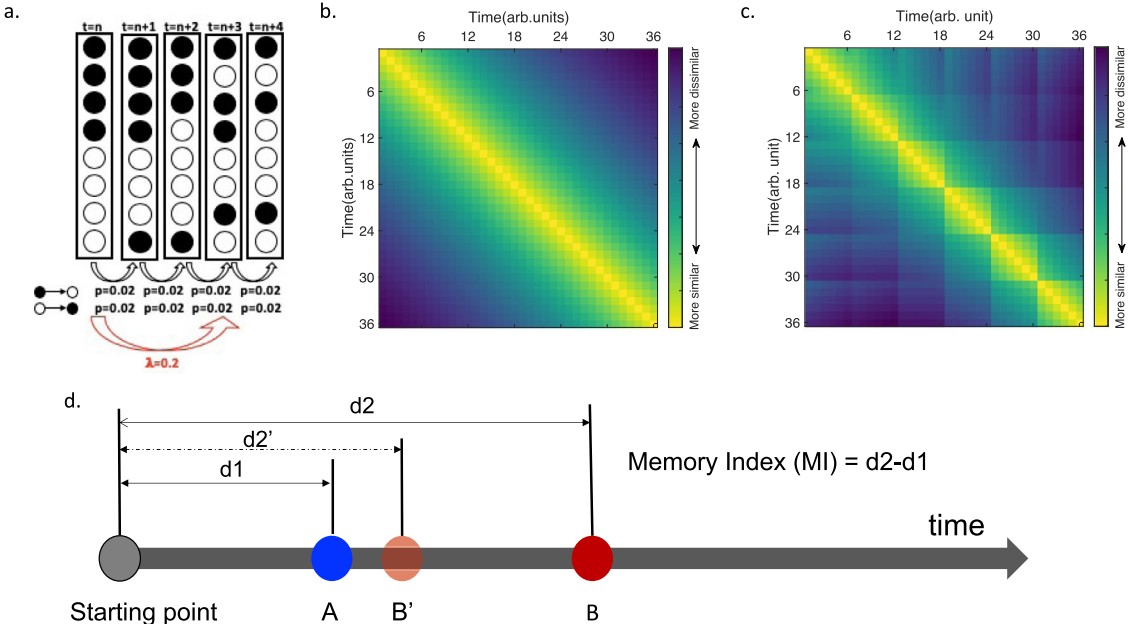

**Fig. 4 Computational model. a** Schematic of the model. The context signal was defined as a vector of binary elements, which drifts at rate p when there is no event boundary and resets at rate λ at event boundaries. **b**, **c** shows representational dissimilarity (1-R) for each time point relative to all the other time points averaged over 1000 iterations of the simulation of the model containing no event boundaries (**b**) and the model containing 5 event boundaries (i.e., event boundary occurred every 6 time points, **c**), respectively. **d** Schematic of the memory index (MI) used to quantify recency judgments. To make a recency judgment, the learning contexts of each probed object is reinstated, and compared to the contextual representation of the first time point during encoding (i.e., d1 and d2, which refer to the representation dissimilarity between each of the probed items and the reference item). The one more dissimilar with the reference point (quantified by d2-d1) is judged as the item presented later. arb. units refers to arbitrary units. Source data are provided as a Source Data file.

random factor to control for a potential clustering effect due to data independency. We repeated the procedure above using a range of parameter values of p (from 0.01 to 0.4, with steps of 0.01) and λ (from 0.05 to 1, with steps of 0.01), and found that our model can explain our behavioural data across a range of parameter values (Fig. 6a, b, see also Supplementary materials (Supplementary Figure 7) for another example of the model fitting with a different set of parameter values), indicating the robustness of the model to parameter values. Note that the model is not insensitive to parameter values, as only a range of parameter values produce model outputs that can both correctly recover all the behavioural effects (indicated by the red line in Fig. 6a, b) and significantly correlate with all the behavioural data across experiments (indicated by the black line in Fig. 6a, b). In Supplementary materials (Supplementary Figure 8), we also show two examples of model simulations which could not account for the behavioural data.

Next, we examined whether using the last contextual representation during encoding as the reference as opposed to the first contextual representation improves the model's accuracy. Results showed that under many parameter value combinations, model outputs significantly correlate with the behavioural data, but cannot correctly recover all the effects in the behavioural experiments (Fig. 6c, d).

We also ran a comparable analysis on the reaction time (RT) data. We found that RT significantly correlates with the memory index (MI) in our model across a range of parameter (p and λ) values (see Supplementary Fig. 19). The correlation pattern across parameter values is similar to the correlation pattern between accuracy and the MI; albeit the $R^2$ value was generally smaller for RT than for accuracy. This result is however not surprising, considering that participants were instructed to prioritize accuracy. As such, the RT measurements are noisier than accuracy to index TOM performance.

Finally, we compared our model to a previous model[5], which postulates a random sharp shift in context signals at event boundaries (e.g., the rate of context change increased from 0.01 to 0.08). Note that in this model, the sharp change at event boundaries is random, in the sense that a random set of elements in the context vector changes (0 will change to 1, 1 will change to 0) at event boundaries. However, the magnitude of the change at event boundaries is not random, but depends on how much change occurs in the internal/external environment[5]. In their model, Horner, Bisby[5] used the contextual similarity of the two probed items as the index for the accuracy of TOM. Simulation results showed that Horner, Bisby[5]' model does not licence the co-existence of the boundary effect and the temporal distance effect (see Methods and Supplementary Fig. 5). Since there is no metric for recency judgments in their model, we combined Horner, Bisby's[5] model with our proposed metric. Simulation results showed that across parameter values, their model explained the data less well than our model (Supplementary Figs. 10 & 11).

Taken together, our computational model explained our findings on TOM and revealed two key points. First, event boundaries do not cause a random sharp change as previously proposed ([5]); Instead, it resets temporal context by recovering a certain proportion of the first contextual representation. Second, the latent cause for better TOM is not increased contextual similarity/dissimilarity of the two probed items themselves, which has been the focus and assumption of previous theories and computational models (e.g.,[5,21,22]). Instead, the latent cause for better TOM is increased magnitude of *d2–d1*, in which d refers to the distance between the context of the probed item and the first context. By combining these two propositions, our model not only resolved the theoretical and computational conundrum of why both contextual similarity and dissimilarity have been observed to promote TOM in previous empirical studies

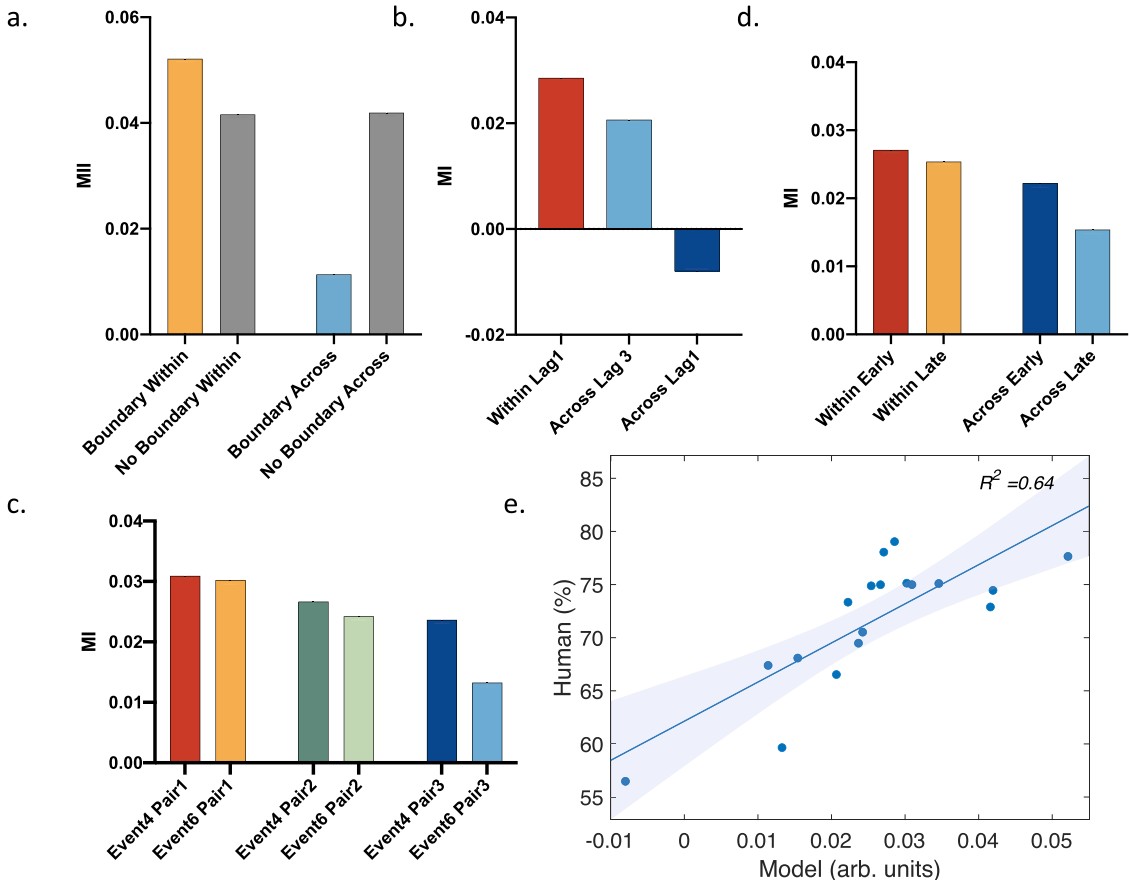

**Fig. 5 Model results. a–d.** Model outputs as quantified by our memory index (MI, i.e., d2-d1 as shown in Fig. 4d.) for Experiments 1-4, when the drift rate and reset rate were set at $p = 0.02$ and $\lambda = 0.2$. **e** Pearson correlation between model outputs shown in Fig. 5a–d and the corresponding group averaged behavioural results pooled together across behavioural experiments. The shaded area represents 95% confidence interval of the regression line. arb. units refers to arbitrary units. Source data are provided as a Source Data file.

(e.g.,[25,33]), but also captured all the other boundary-related effects found in the present study.

## Discussion

The present study investigated how and why event boundaries structure the temporal organization of memory. Across four empirical experiments, we unraveled a number of behavioural phenomena: event boundaries exert a dual role, both enhancing and impairing TOM for within and across-event TOM respectively; event boundaries cause an advantage in TOM for earlier local event positions (the local primacy effect), such that decreasing event length by increasing the number of event boundaries in a list enhances TOM for both within and across-event pairs at earlier event positions, even when controlling for the absolute list positions; the boundary effect can co-occur with the temporal distance effect, although they were thought to be computationally incompatible. We then developed a parsimonious algorithmic model to mechanistically account for all these results. Our model shows that event boundaries reset temporal context signal, and the latent cause for better TOM is increased difference between the contextual representations of the probed items relative to the first contextual representation (i.e., increased magnitude of d2–d1, Fig. 4d). Our model thus provides a unified mechanistic account of how organizational factors (e.g., temporal distance, context) during encoding affect TOM.

Based on the Event Horizon Model[10], we reasoned that event boundaries might benefit within-event TOM, since event boundaries should decrease competition among items chunked in the same event. However, event boundaries might also increase the competition among different events, and therefore might impair across-event TOM. Our data showed that compared to the no boundary condition, TOM for within-event pairs was enhanced and TOM for across-event pairs was impaired in the boundary condition, pointing to a dual role of event boundaries in structuring TOM. These results also provide behavioural evidence for the idea that event boundaries promote integration for items in the same event and separation in different events (see[34] for a review). We further investigated how TOM is affected if event length was decreased by adding more event boundaries in a list. To our surprise, decreasing event size does not cause a generic TOM improvement for within-event pairs and impairment for across-event pairs, as predicted based on the idea of decreased competition among items and increased competition among events in a list chunked by more event boundaries[10]. Instead, we found that TOM was comparable if list positions and local event positions were identical; and was enhanced both for items within and across-event pairs that took earlier local event positions in shorter events, relative to items that took later local event positions in longer events. We termed this phenomenon the local primacy effect. The local primacy effect differs from previous reports of memory improvement when introducing more event boundaries[6,35–37]. The local primacy effect describes a phenomenon in which improvements in memory are strongest at the beginning of an event and gradually decrease as event positions move away from the event boundary. In contrast, the memory improvements reported in previous studies are either

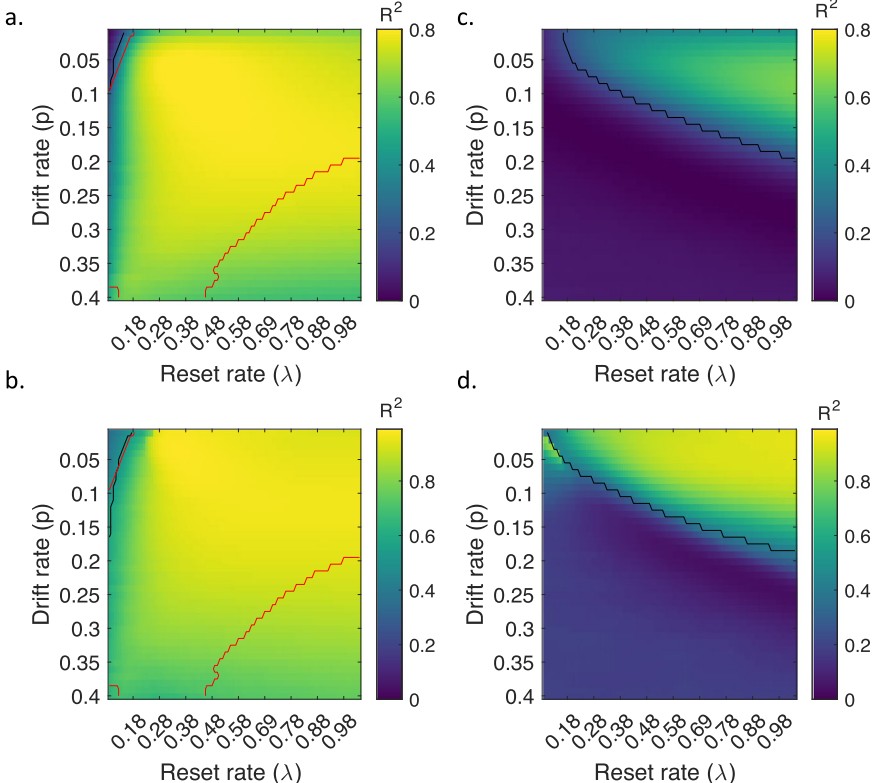

**Fig. 6 Correlation between model outputs and the behavioural results under different parameter values. a** Explained variance ($R^2$) of the behavioural results by the model outputs, fitted with a generalized linear model. **b** Explained variance ($R^2$) of the behavioural results by the model outputs, fitted with a generalized linear mixed model. In **a**, **b**, the first contextual signal was used as the reference point for recency judgments. **c**, **d** are identical to **a**, **b**, except that the last contextual signal was used as the reference point for recency judgments. The black line in images indicates the significance threshold of $p = 0.05$, and the red line indicates the parameter values that yield model outputs which can recover all the effects in the four behavioural experiments. Source data are provided as a Source Data file.

restricted to event boundaries or show no gradual change as a function of the proximity to the event boundary[6,35–37]. In addition, those memory improvements are in the domain of non-temporal memory, such as an increase in item memory or source memory[6,38], whereas the local primacy effect reported here refers to TOM. Overall, these results demonstrate complex scenarios of how event boundaries affect TOM than previously hypothesized, and therefore pose a new challenge to theoretical and computational models.

Moreover, the rebound in TOM for items across an event that are separated by more vs. fewer intervening items replicates the well-established temporal distance effect, while also helps rule out confounds as the root of impaired performance for across-event pairs, e.g., participants might simply not link different events at all. If participants did not try to remember the order of across-event items as instructed, this would lead to a generic low performance (e.g., chance level) for all across-event pairs regardless of the number of intervening items.

Our algorithmic model could account for all behavioural effects, including the boundary effect and the temporal distance effect, two effects seemingly computationally incompatible, as well as the local primacy effect observed in the present study. In our model, in the absence of an event boundary, context signals drift gradually over time. At event boundaries, context signals shift sharply via a resetting process, whereby a certain proportion of the first context signal, which contains both pre-experimental context and the first context in a list is reinstated. To operationalize recency judgments, we developed a metric, i.e., during retrieval, the learning contexts associated with the two objects are reinstated[32] and compared to the absolute first

contextual representation during encoding. The item more/less similar to the first contextual representation (quantified by $d2-d1$, Fig. 4d) is judged as the one presented earlier/later[24]. This model reveals key insights that allow accounting for all the effects. One of those insights is that the resetting process at event boundaries increases the difference between the contextual representations of the probed items relative to that of the first contextual representation (i.e., larger magnitude of $d2-d1$) across multiple conditions: for within vs. across-event pairs, for pairs separated by more vs. fewer intervening items, as well as for pairs at earlier vs. later local event positions. It might be less intuitive to understand how the resetting mechanism at event boundary increases the magnitude of $d2-d1$ for earlier vs. later local event positions, given the constant drift rate. This is because while the drift rate, which defines the rate of change between one context and its neighbours, is constant, the amount of change between two neighbouring contexts relative to the first context (i.e., $\Delta d = d_{t+1}-d_t$) is not constant across positions. In fact, the more dissimilar the contextual representation to the first one is, the smaller $\Delta d$ becomes. This is because reinstating a certain proportion of the first context at event boundaries makes the contextual representation of each event boundary more similar to the first context compared to the contextual representations of pre-boundary items, leading to larger $\Delta d$ between event boundary and its next item. Hence, the accumulated $\Delta d$ between two items (e.g., the accumulated $\Delta d$ between item $t+2$ and item $t$ is $(d_{t+2} - d_{t+1}) + (d_{t+1} - d_t) = d_{t+2}-d_t$) is larger for earlier event positions versus later event positions. Since the accumulated $\Delta d$ between two items is the contextual difference of the two probed items relative to the first one (i.e., $d2-d1$ in

Fig. 4d, see above the example formulas on why $\Delta d$ between item $t + 2$ and item $t$ is $d_{t+2}-d_t$), this explains why our model can account for the local primacy effect.

Our model also predicts a primacy effect in the long list i.e., better TOM for early than late list positions), since $\Delta d$ is larger for earlier positions than for later positions (see why this is the case in the previous paragraph). Our model further predicts that the magnitude of the decrease in TOM accuracy should be larger for the no boundary condition than for the boundary condition. This is because in the boundary condition, the resetting process at event boundaries increases $\Delta d$. To test this prediction, we calculated the average TOM for early and late [matched] within-event TOM (see Supplementary Fig. 3a) for both boundary condition and no boundary condition. We then ran a two (Conditions: boundary condition vs. no boundary condition) by two (Positions: early vs. late) repeated measures ANOVA. Following the model predictions, we expected a Condition by Position interaction. In line with this prediction, a significant Condition by Position interaction was found ($F_{(1, 25)} = 4.601$, $p = 0.0419$, see Supplementary Fig. 3b). That is, in both conditions, there was a decrease in TOM accuracy from early to late positions (early vs. late position in the boundary condition: $t(25) = 2.392$, $p = 0.0246$, two-sided and in the no boundary condition: $t(25) = 5.425$, $p < 0.001$, two-sided). Critically, and consistent with our model predictions, TOM was significantly worse for late position in the no boundary condition compared to in the boundary condition (boundary vs. no boundary for late position: $t(25) = 3.156$, $p = 0.0041$, while $t(25) = 0.1222$, $p = 0.9037$, two-sided for early position).

Taken together, our model provides a unified solution to explain a range of effects found in the present study. It demonstrates that the latent cause for better TOM is the increased magnitude of $d2-d1$, instead of the contextual similarity or dissimilarity of two probed items (see Supplementary Fig. 6 for a demonstration that contextual similarity fails to explain the local primacy effect). The latter is often an assumption made by previous models and the focus of many neuroimaging studies, which inevitably leads to paradoxical observations in empirical studies (e.g., sometimes contextual similarity promotes TOM and sometimes contextual dissimilarity promotes TOM[23]). Therefore, our model resolves the long-standing paradox in memory literature on TOM, and provides a unified algorithm to mechanistically account for how organizational processes during encoding influence TOM.

Notably, our simulation results showed that substituting the resetting process with a random sharp shift as proposed in Horner, Bisby[5]'s model, substantially decreased the models' ability to explain the empirical data. This result gives independent support to the hypothesis that event boundaries do not cause a random change, and instead a resetting process might underlie how event boundaries affect the temporal organization of memory. As a consequence of such a resetting process, one important, yet untested, prediction from our model is that boundary items will become more similar to the first items in a sequence containing event boundaries vs. not containing event boundaries. This testable prediction is worth investigating in future studies.

Moreover, there remains a key question for future studies. That is, what information is contained in the beginning to be carried over to each of the following events. We speculate that there must be some abstract, schematic and generalizable information that is activated in the beginning and recovered at the event boundaries. Recent work based on computational modeling and recordings from rodents during spatial navigation[39] suggests that structural information is preserved across boundaries. It remains to be tested whether this is also the case in humans with an appropriate experimental design. What's more, it worth noting that one

limitation of the current study is that the structure of each individual event is identical (e.g., every event contains six sequentially presented items sequentially). It should be tested whether similar effects found in the present study will also be found when the structure of each event in the long list is not the same.

We acknowledge other classes of models. However, those are challenged by our behavioural results. For instance, rehearsal-based models (e.g.,[40–42]) assume that associations are formed directly between items that are rehearsed successively, with association strength increasing as a function of the number of rehearsals. This model could not explain why TOM is better for items taking earlier vs. later local positions, since the number of rehearsals those item pairs might receive is identical (e.g., items 2–4 in the Event 4 condition vs. items 4–6 in the Event 6 condition in Experiment 4, Fig. 3a). Moreover, this model would even predict that items 2–4 in the Event 6 condition are remembered better than items 2–4 in the Event 4 condition, as there should be more rehearsals for items 2–4 in an event with more items. A prediction that runs contrary to our empirical data. Another influential model, the Event Horizon Model ([10]), which attributes the benefits of event segmentation to reduced interference also does not easily explain the increase in TOM for across-event pairs taking earlier vs. later positions. This is because interference is comparable for within and across local events when the number of items in local events and the number of total events in the long list are identical (e.g., across-event pairs in Experiment 4).

Other studies (e.g.,[43]) have proposed that recency judgments may rest on item strength, such that items with stronger memory are judged as more recent. However, evidence supporting such an item strength-based mechanism (ISBM) in recency judgments is mixed (e.g.,[32,43–46]). More critically, predictions based on ISBM run contrary to our empirical observations. For instance, ISBM would predict better memory for Across Lag1 than for Across Lag3 and Within Lag1 in Experiment 2. This would be expected, since Across Lag1 includes probed pairs spanning boundary items, which are known to be remembered better and thus have stronger memory strength (see[34] for a review), and also happen to be the more recent item.

Previous studies have shown that event boundaries may also alter perceived temporal distance between two items (e.g.,[47]). Naturally, extending our model to account for such a related phenomenon would be a next step. Our model uses $d2-d1$ as the metric for temporal order memory, assuming that a reference point is necessary to make recency judgments. Assuming that temporal distance judgments are based on a similar decision-making process, our model would predict longer perceived temporal distance for within-event than across-event pairs. However, judgments of temporal distance might not need to rely on a reference point, but instead may rely on the contextual similarity of the two probed items. Here, two items with higher contextual similarity would be judged as closer together than items with lower contextual similarity. If so, perceived temporal distance would be expected to be shorter for within-event than across-event pairs. Results from empirical studies (e.g.,[21,47,48]) support both contradictory predictions, stressing the need for further studies to fully understand how event boundaries affect temporal distance estimates.

In conclusion, the present study reveals behavioural phenomena of how event boundaries structure the temporal organization of memory: It provides behavioural evidence for the dual-role hypothesis of event boundaries in affecting TOM, and reveals a primacy effect caused by event boundaries, explaining why varying event length by varying the number of event boundaries in a fixed-length list affects TOM. By proposing a reset of temporal context signals at event boundaries, we provide a unified

algorithmic mechanism for understanding how organizational processes during encoding influence TOM. This model not only reconciles a long-standing paradox of why sometimes contextual similarity and sometime contextual dissimilarity promotes TOM, but also makes testable predictions for future studies.

## Methods

**Experimental paradigm across experiments**. The paradigm used in present study was adapted from previous studies[6,7]. In this task, participants remembered lists of grey scaled pictures. Each list contained 36 trial-unique images, sequentially presented to participants. The object images used in the present study were taken from the bank of the standardized stimuli ([49,50]), which were normalized for name, category, familiarity, visual complexity, object agreement, viewpoint agreement, and manipulability. Each image was resized to 350 by 350 pixels. Images were embedded on a coloured frame (e.g., Fig. 1a), whose colour switched after a certain number of objects depending on the specific experimental manipulation. Frame colours did not repeat within one sequence and were distinguishable.

Each object image was presented on the screen for 2.5 s, followed by 2 s' inter-trial interval (ITI) where the colour frame remained on the screen. Following the ITI, a fixation cross embedded on the colour frame was presented for 0.5 s before presentation of the next object. At boundary trials, the colour of the frame updated with the co-occurring object. Event boundaries were operationalized as such perceptual shifts.

The task consisted of an encoding and a retrieval phase. In the encoding phase, participants were instructed to imagine the object in the colour of the frame, and quickly indicate their liking for such a combination by pressing a button during the presentation of each picture. In the meantime, they also needed to remember the order of all the 36 pictures. To promote memory performance, they were encouraged to imagine vividly neighbouring items interacting with each other regardless of the colour change (e.g.,[51]). In the retrieval phase, participants made self-paced recency judgments on pairs of objects chosen from the just-encoded sequence. They were instructed to respond as accurately as possible (i.e., prioritize decision accuracy over reaction time), and once they had decided which picture to choose, they pressed the button as fast as possible. Each object appeared only once during the test trials. Each recency judgment trial was followed by a confidence rating, in which participants rated their previous decision on a four-point scale (1 = low confidence and 4 = high confidence). The object pairs from the first half of the learned sequence were tested first. After finishing the encoding and testing of each sequence, participants took a short self-paced break.

**Participants recruitment**. The participants were recruited from Frankfurt am Main and its neighbouring areas. All participants had normal-to-corrected-to-normal vision and normal colour perception. They had no past or present psychiatric disorders, and did not take any psychoactive or hormonal medications at the time of testing by self-report. All experiments were approved by the Ethics Council of Max Planck Society. All participants gave their written informed consent and were compensated financially for their participation. A minimal sample size of 26 for each experiment was decided based on a power analysis (significance level 0.05, statistical power 0.80) of a previous paper ([7]) on a similar boundary-related memory effect (i.e., better TOM for within vs. across-event items).

### Experiment 1

*Participants*. 26 right-handed male and female naive participants (females = 18; mean age = 25 years; range = 20–33 years) participated in Experiment 1. Two additional participants were excluded from data analyses because they did not finish the task.

*Experimental design and procedure*. Participants encoded and were tested on 14 sequences. Half of the sequences contained a frame whose colour changed after every 6 images (i.e, boundary condition) and half of the sequences contained a frame whose colour was kept constant across all 36 images in one sequence (i.e., no boundary condition), but changed across different sequences. 24 unique colours were used in the boundary condition and were recycled after every four sequences. 7 colours from a subset of the 24 colours were used in the no boundary condition. Two random sequences of object images were generated for each condition and randomly assigned to each condition. The sequences of object images were counterbalanced between conditions across participants.

In the testing phase, participants made recency judgments on two pair types in the boundary condition, i.e., within-event pairs and across-event pairs (Fig. 1b). In the no boundary condition, we tested pairs taking identical list positions as in the boundary condition. Although there was no within vs. across-event difference between those pairs in the no boundary condition, for convenience, we named those pairs as matched within-event or matched across-event pairs according to their corresponding list positions in the boundary condition. In total, there were 7 sequences × 6 pairs = 42 within-event pairs and 7 sequences × 5 pairs = 35 across-context pairs in each condition.

The experiment was a two by two within-subject design, with the factors being Condition (boundary vs. no boundary condition) and List Position ([matched] within vs. [matched] across-event pairs). The two conditions were presented to participants in blocks of 3–4 sequences in an interleaved fashion and were counterbalanced across participants. Participants practiced on two additional sequences (one for each condition) to familiarize themselves with the task before the main experiment. All experiments reported were controlled via Presentation software on a Fujitsu Celsius M730 computer running on Windows 7 (64 bit). The experiment was carried out in a dimly lit, soundproof booth. Due to a technical problem, only 12 sequences were recorded in one participant and 13 sequences in two participants.

### Experiment 2

*Participants*. A total of 27 right-handed male and female naive participants (females = 18; mean age = 24 years; range = 18–32 years) participated in Experiment 2.

*Experimental design and procedures*. Participants encoded and were tested on 14 sequences of 36 images and practiced on two additional sequences before the main experiment. We generated two random orders for the images presented in the 14 sequences in the main experiment and randomly assigned them to participants.

Three types of item pairs were tested after encoding each sequence, i.e., within-event pairs (items 2–4) with one intervening item (within lag 1), across-event pairs (items 3'-1) with one intervening item (across lag 1) and across-event pairs (items 3'-3) with three intervening items (across lag3, see Fig. 2c). In total, there were 14 sequences × 9 pairs = 126 within-event pairs, 14 sequences × 8 pairs = 112 across-event pairs with each half (i.e., 112/2 pairs) containing pairs separated by one intervening item and three intervening items respectively.

The experiment was a one-way within-subject design, with the factor being pair types (within-event pairs at lag 1 vs. across-event pairs at lag 1 vs. across-event pairs at lag 3).

### Experiment 3

*Participants*. A total of 31 right-handed male and female naive participants (females = 18; mean age = 26 years; range = 20–35 years) participated in Experiment 3. Data from two additional participants were not recorded due to a technical problem.

*Experimental design and procedures*. Participants were exposed to 14 sequences of images. In half of the sequences (7), the colour of the frame changed after every 4 images (the Event 4 condition). In the other half (7), the colour of the frame changed after every 6 images (the Event 6 condition) (Fig. 3a). Due to a technical problem, only 13 sequences were recorded in one participant and 8 sequences in another participant.

Three pair types were tested in the two conditions (Event 4 vs. Event 6, Fig. 3a). Pair type 1 was within-event pairs, taking identical list positions and identical local event positions in both conditions. Pair type 2 was also within-event pairs, which took identical list positions, but earlier local positions in the Event 4 condition and later local positions for the Event 6 condition. Pair type 3 was across-event pairs. Due to different number of event boundaries in the two conditions, they could only be matched for the average list positions in the two conditions, and the local positions were always earlier in the Event 4 condition than the Event 6 condition. In total, there were 7 sequences × 3 pairs = 21 pairs for pair type 1 and 2 respectively, and 7 sequences × 4 pairs = 28 pairs for pair type 3 in each condition. The two conditions were presented to participants in blocks of 3–4 sequences in an interleaved fashion. The order of the two conditions were counterbalanced across participants.

The experiment was a two by two within-subject design, with the factors being conditions (Event 4 vs. Event 6) and pair types (pair type 1 vs. 2 vs. 3).

### Experiment 4

*Participants*. 30 right-handed male and female naive participants (females = 22; mean age = 24 years; range = 18–30 years) participated in Experiment 4. Data from two additional participants were excluded from data analyses since they did not finish the task due to drowsiness.

*Experimental design and procedures*. Participants were exposed to 14 sequences of images. 36 unique RGB colours were chosen for the background frames, and were recycled after every four sequences. We created two types of lists, in which patterns of the colour change were 3-3-6-3-3-6-3-3-6 (for 7 sequences) and 6-3-3-6-3-3-6-3-3 (for the other 7 sequences), meaning that the colour of the frame was constant for either 3 or 6 objects (see Fig. 3c). The total number of event boundaries was identical in the two types of lists. The frame colour used in the two list types were counterbalanced across participants.

Three types of pairs were tested for recency judgments: within-event pairs, across-event pairs of a shorter lag and across-event pairs of a longer lag (Fig. 3c). The within-event pairs and across-event pairs of a shorter lag were the critical testing pairs for examining the hypothesis in the present experiment. Each pair type consisted of two sub-types, i.e., earlier and later pairs, with the ordinal

positions of the two sub-types in the long list being kept identical. In total, there were 42 within-event pairs taking earlier and later event positions respectively (marked by red and yellow squared brackets), 35 across-event pairs of a shorter lag taking earlier and later event positions respectively (marked by dark blue and light blue squared brackets respectively), and 70 across-event pairs of a longer lag (marked by grey squared brackets). Due to a technical problem, only 13 sequences were recorded in three participants.

*Computational modelling.* We built our model based on a class of temporal context models (e.g.,[12,13]), in which items are associated with each other via a context signal. Mathematically, context was defined as a set of binary elements with 100 features. At non-boundary time point, context fluctuates from moment to moment, with active elements (1) turning off and inactive elements (0) turning on with probability $p$, such that

$$C_t = (1-p)C_{t-1} + pC^{IN} \quad (3)$$

where $C_t$ represents context representation at time point t, $C^{IN}$ represents random noise. When a boundary occurs, context signal resets at rate $\lambda$, such that

$$C_t = (1-\lambda)((1-p)C_{t-1} + pC^{IN}) + \lambda C_1 \quad (4)$$

We simulated 36 time points, matching the object number in our experiments. Event boundaries occurred after every certain number of time points according to specific experimental design. The similarity between two context signals was quantified using Dice similarity coefficient. We developed a metric, termed memory index (MI) to quantify recency judgments (Fig. 4d). $MI = (1-R_2) - (1-R_1)$, where $R_2$ refers to the similarity of the context signals between the more recent item and the first item, and $R_1$ refers to the similarity of the context signals between the more distant item and the first item. The first contextual representation was used as a strategical reference point, in a similar fashion adopted in[52]). $1-R_2$ and $1-R_1$ are referred to as $d2$ and $d1$ in Fig. 4d respectively.

Previous studies (e.g.,[5,21]) used contextual similarity of the probed items as a proxy for temporal order judgments. However, considering only the context similarity of the probed items allow neither temporal order judgments of the two items, nor the coexistence of the boundary effect and temporal distance effect at the computational level (see Supplementary Fig. 5 for simulation results based on Horner, Bisby[5]'s model). Moreover, we did not implement a scanning mechanism in our metric, as adopted in previous models (e.g.,[46]) to account for internal time estimate and recency judgments. This is because in our task we did not find clear behavioural evidence supporting a backward or forward sequential scanning process, i.e., we did not find that reaction time of recency judgements decreases as a function of the proximity of the most recent item in the probed pairs of items to the last item in the list (Supplementary Fig. 4), as shown in some earlier studies[53–55]. Since these earlier studies are more likely to tap into short-term memory, the discrepancy between our study and these earlier studies might be due to differential retrieval mechanisms in short and long-term memory. There is no clear evidence supporting sequential scanning in most tasks tapping into long-term memory (see[56] for a review). In addition, DuBrow and Davachi[25] suggest that intervening items between two probed might be sequentially activated during recency judgments. However, closer inspection of their decoding evidence based on the functional magnetic resonance imaging (fMRI) data revealed that such a process is most apparent only in no switch condition (equivalent to within-event pairs in our experiment); No conclusive evidence was found in switch condition. Moreover, no clear evidence based on their behavioural data (e.g., reaction time) supporting sequential scanning. That is, they did not find a significant difference in reaction time during recency judgments between long lag pairs and short lag pairs. In our data, we did not find evidence supporting such a process either. For instance, reaction time was not significantly longer for Across Lag3 than for Across Lag1 (see Supplementary Fig. 1b). Therefore, we did not include a sequential activation process in recency judgments in our model.

We ran 1000 differently seeded iterations of the model according to our experimental manipulations under a range of parameter values of drift rate $p = 0.01$ to 0.4 (step size of 0.01) and $\lambda = 0.05$ to 1 (step size 0.01). We then fit a generalized linear model (GLM) and a generalized linear mixed model (GLMM, in which, experiments/sequences were set as a random factor to control for a potential clustering effect due to dependency of group averaged behavioural data from the same experiment/sequence) to examine whether our model outputs could still significantly explain our behavioural data.

We also applied a similar analysis for reaction time (RT) as for accuracy above, to examine whether our model outputs could significantly explain RT. Note that our task was not setup for participants to respond as accurately and as fast as possible, but instead to prioritize accuracy at the expense of speed (see Methods). Although RT might be much nosier than accuracy to reflect TOM performance, it might be able to reflect the difficulty of TOM judgment. Therefore, we examined whether the model outputs negatively correlated with RT.

Finally, we compared our model to a previous model[5]. Horner, Bisby[5]'s model implemented a random sharp shift in temporal context at event boundaries, e.g., context change rate increased from 0.01 to 0.08 at boundaries, but not a resetting process. As shown in Supplementary Fig. 5, their model could not recover both the boundary effect and the temporal distance effect, partly because the metric they used to index the accuracy of TOM failed to do so. We therefore combined Horner, Bisby[5]'s model with our proposed metric (i.e., d2–d1 for recency judgment index),

to estimate how well the outputs of their model explained the behavioural data, fitted with both the GLM and the GLMM.

*Statistical analysis.* To test for the statistical significance of the results from empirical experiments 1–4, ANOVA followed by false discovery rate (FDR) multiple comparison tests was performed in GraphPad Prism 8. The data were first tested for equal variance of differences. If the data do not meet equal variance, the Geisser-Greenhouse was used for correction. In the computational modelling section, fitting a generalized linear model (GLM) and a generalized linear mixed model (GLMM) was performed in matlab (Mathwork R2019b).

**Reporting summary.** Further information on research design is available in the Nature Research Reporting Summary linked to this article.

## Data availability
The processed data for the analyses and underlying figures have been deposited in an Open Science Framework database (https://doi.org/10.17605/OSF.IO/UK4DH), and can be found in the Source data file that is provided with this paper. Raw data cannot be shared publicly, as informed consent for data sharing was not obtained from participants at the time of data collection. Interested researchers may request raw data and obtain a de-identified, minimal dataset pending ethical approval from the authors' institute. Data requests can be sent to Dr. Yi Pu (yi.pu@ae.mpg.de). Source data are provided with this paper.

## Code availability
The Matlab code for simulation is available from the following link: https://doi.org/10.17605/OSF.IO/NW36Q.

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

## Acknowledgements
We thank Dr. Caspar Schwiedrzik for proofreading and commenting on our manuscript. This work was supported by the Max Planck Society. C.R. and L.M. were supported by Multi-University Research Initiative Grant (ONR/DoD N00014-17-1-2961), X.Z.K. was supported by the Fundamental Research Funds for the Central Universities (2021XZZX006), the National Natural Science Foundation of China (32171031), and Information Technology Center, Zhejiang University.

## Author contributions
Y.P., L.M. conceived the project. Y.P. designed the experiments. Y.P. performed the empirical experiments and analyzed the data. Y.P., X.Z.K. performed the computational modelling. All the authors discussed the data and provided critical feedbacks. Y.P. wrote the original draft. Y.P., X.Z.K., C.R., L.M. reviewed and edited the draft. L.M., C.R. supervised the project.

## Funding

## Competing interests
The authors declare no competing interests.
