## [Peer Review File · Nature Communications]

Event boundaries shape temporal organization of memory by resetting temporal contextREVIEWER COMMENTS

Reviewer #1 (Remarks to the Author):

The paper provides strong evidence that event boundaries both reduce TOM across events and enhance TOM within events. The authors further show that this latter benefit may be due to a “local primacy effect” whereby items presented early on in an event are better remembered than items later in an event, even when testing across-event memory. Importantly, they propose a new model whereby boundaries are associated with a reset that recovers a “beginning” context that drifts slowly across the event. This set of findings and the proposed model reflect an important theoretical contribution towards understanding how event segmentation may improve memory, suggesting there are lingering benefits after an event boundary related to contextual resetting that fade over time. My major concerns, described below, are mainly related to some interpretation of the empirical data, to the possible high sensitivity of the model to event position, and to the consideration of the prior literature.

Empirical data

In general, the authors do an excellent job controlling for possible confounds. However, there are a few places I am concerned about list position confounds in particular. In experiment 1, the use of a no boundary list is an excellent control, however boundary within and boundary across are still directly compared. Given that task diagram shows that the within event condition contains items from more primacy and recency positions in the list, I would recommend that comparison be removed. Alternatively, the authors could rerun the analysis excluding the first and last within event condition, which would presumably bias the effect in the reverse direction. Again, in experiment 2, the positions of the within versus across lag 1 appear biased to show the boundary effect on TOM. That is, the within lag 1 condition occupies more primacy and recency positions than both across-event conditions. Subsampling may be necessary to address this issue.

Again in experiment 3, there is commendable effort put into matching the list positions of the within-event conditions that cleverly allows comparisons of early versus late event effects. However, the across event condition with shorter events/earlier event positions (3&3 for event 4 list vs 3&5 for event 6 list) is biased to show a short event/early event position enhancement because those also occupy more primacy and recency positions in the list (matching on average position doesn't help since primacy and recency counteract, even though they should both enhance memory). This is not a fatal flaw, as the next experiment matches the positions, but I think the authors should not report that effect in experiment 3 without significant qualifiers describing this confound (otherwise, again, subsampling could help).

In experiment 4, early versus late comparisons are perfectly matched, but the temporal distance effect pairs are not (by definition). The problem is, the longer pairs in grey include really early list items (the 2nd) which may be very easy to remember, while the shorter pairs do not contain any primacy or recency items within three positions from the start or end of the list. The authors should again subsample, dropping these first long-item pairs and/or qualify their description of the distance effect based on this potential confound.

Model

The proposed model is appealing in that it reconciles boundary and distance effects on TOM, but a have a couple concerns. First, it is somewhat bewildering that the model is so insensitive to parameter values. It would be important to show that at extreme values (e.g., very low reset,) the fit drops.

Critically, it seems that the within versus across effect would be very sensitive to the actual event positions sampled. If you are computing d_2-d_1 on position 2 versus 3, that will be much more difficult a judgment than 2 versus 5. So it seems like performance might be less sensitive to there being a

restart at the boundary as it is to the exact positions sampled for testing. This would explain why the simulation recovers the across early versus late effect, since the positions sampled are closer in the late versus early condition (note, fig 5D is mislabeled “across early” in both). Additional simulations should be included to address these concerns (e.g., comparing within versus across while matching the absolute value of $d_2 - d_1$). It would also be helpful to show whether and why the model predicts the local primacy effect given that the drift rate is constant.

Although the model explains TOM, it cannot account for the prominent related effect that boundaries increase perceived temporal distance (e.g., as shown in the Ezzyat and Rouhani references as well as others). Indeed, it seems like it would make the opposite prediction if using the $d_2 - d_1$ metric. If the model can account for it, the authors should include relevant simulations, as that would be a significant contribution. Otherwise, some discussion of this limitation should be included as it is a highly relevant effect in the literature.

Literature

When citing a relatively new finding like the boundary effect on TOM, the first paper(s) to show it should be referenced. The first, as far as I know, was DuBrow & Davachi (2013). This paper was cited in the methods for the design but not for its theoretical contribution. Moreover, the paper reports both a boundary effect on TOM (reduced) and a distance effect on TOM (enhanced) that the current manuscript suggests had not previously been demonstrated to “co-exist in the same experiment.”

The current manuscript cites Jenkins and Ranganath (2016) for the “temporal distance effect.” That paper only probes a single temporal distance so it is impossible to show the distance effect. The authors should reference papers that do show the effect, e.g., originally Ynetma & Trask (1963). For a more recent demonstration, see St Jacques et al (2008). They should also provide references for this statement “hippocampal representations are more dissimilar for experiences further apart than experiences close in time” and consider drawing on the animal literature.

In the next paragraph, Ezzyat and Davachi (2014) is somewhat misrepresented, as essentially the same number of regions showed within-event similarity as across-event similarity. Later in the manuscript (pg 14) it is referenced along with the Jenkins paper and another preprint as showing more similarity between item pairs with shorter vs longer lags, but neither of the peer-reviewed papers show that effect. Indeed, they don’t even manipulate lag, and, even if they did, that would be meaningless due to the slow drift of the BOLD signal. Later again, in the discussion, the Ezzyat paper is referenced as showing TOM, which it did not test.

The authors reference the models in Horner et al (2016) and Rouhani et al (2020) as accounting for “either the ‘temporal distance’ or the ‘boundary effect’” but I don’t see anything in either model regarding the temporal distance effect.

At least two additional relevant models should be referenced:

Howard et al (2015, Psych Review) which explicitly modeled TOM.

Henson (1998, Cog Psych) which represents items in terms of their distance from the starting point in a conceptually similar way to present model’s recency judgment metric.

A quick submission of the reference list to this field-specific gender citation balance tool (<https://postlab.psych.wisc.edu/gcbialyzer/>) showed a strong bias in favor of male lead authors. I would strongly recommend expanding the references to include more contributions from the many women authors in this subfield.

Reviewer #2 (Remarks to the Author):

Pu et al present four behavioral experiments and a computational model addressing the role of event boundaries on temporal order memory (TOM). The authors present subjects with a series of objects

temporally separated by color event boundaries, and manipulate the number of events and inter-item lag for TOM judgments. The authors show that boundaries increase TOM within events and decrease TOM across events. They then show that manipulating event length leads to local primacy and demonstrate a boundary effect and temporal distance effect in the same paradigm. The data can be explained by a novel temporal context style model in which the first list context is reinstated at every event boundary.

There are several interesting findings here and the authors have conducted a nice series of well-controlled experiments. My main concerns relate to the model, which is a significant part of the novelty and impact of the manuscript, but has shortcomings that need to be addressed.

Major Comments

1) The authors create a novel computational model that explains the boundary effect and the temporal distance effect, with the main innovation being to have the first context reinstated at each event boundary. My first comment is about this mechanism/assumption: is there any prior behavioral or neural evidence that this is actually what people do at event boundaries that would motivate including it as a mechanism in the model, other than the fits to their own data? As a potential explanation for organization in memory, it seems much more plausible and general that context would simply shift more at event boundaries (Horner et al 2016; DuBrow et al 2017) than be reinstated from an arbitrarily defined timepoint at the start of a list.

2) Related to the previous point, the model fits the data across a wide range of parameter values, but the values that are highlighted $p = 0.02$ and $\lambda = 0.2$. This tells us that the degree to which context moves from moment to moment is ten times smaller than the degree to which context at every event boundary moves back towards the context that existed for the first list item. This seems unlikely to be true in reality, but if the authors disagree they should motivate this result more fully. I also think it would be helpful to present fits to the data (Figure 5) using other parameter values.

3) The authors find in Exp 2 that Within Lag1 performance is better than Across Lag 1 performance and, more surprisingly, Across Lag 3 performance. This is one of the key findings that motivates their argument against a direct associative chaining or similarity mechanism. However, the authors do not consider the potential for some kind of strength based process (e.g. Jenkins & Ranganath 2010) to be at play. The task instructed participants to directly link items in the same event, which could have led to weakened associations between the last item of one event and the boundary item of the next event. This would have produced the Within Lag1 and Across Lag 1 difference that they found. Introducing an additional item strength mechanism for the longer lags could potentially support the intermediate performance in the Across Lag 3 condition. The authors should compare an alternative model that includes such a mechanism, for which there is prior data as motivation.

4) The authors report collecting confidence judgments on the TOM judgment but then do not discuss them. Were the boundary and lag conditions matched in terms of confidence? Knowing this would help answer the question of whether another mechanism, such as a strength process or associative reinstatement, is influencing recency judgments.

5) The authors don't account for previous data showing evidence for reinstatement of direct associations during recency judgments (e.g. DuBrow & Davachi 2014), only mentioning in the Discussion that they did not observe evidence of such a process in their data. While they do show response times in Figure S1 sorted by the position of the first item in a pair, what they need to show is RTs as a function of inter-item distance and boundary condition (i.e. RTs to match the analyses in Figures 1, 2, and 3). However, even if they do not find evidence for inter-item reactivation in response times, they should still explain a more plausible explanation for why their data differ from prior work.

6) There are many places where sentences and paragraphs are more complicated and/or less clear than they need to be, and this definitely affects the impact of the paper. Generally, I try not to comment in reviews on elements of style, but in this case I really think the writing is getting in the way

of the message. For example, in the Introduction, the authors state: "Thus, the basic aspects of how events boundaries structure temporal memories remain to be understood. For instance, whether boundaries enhance and/or impair TOM for items within and across events, respectively...". The first time I read this I was confused because it seems to suggest that prior work has not examined within and across event temporal order memory, which is not the case (and is not actually what the authors mean). The fact that the authors will investigate a novel comparison of a boundary condition with a no-boundary baseline (Exp 1) is not as clear as it needs to be--this is an important aspect of the paper that is confused by the construction of these sentences and the paragraph they are in. This problem exists throughout the manuscript--I think the authors should undertake a thorough revision of the writing, in order to make the findings and their impact clearer.

Minor Comments

1) What exactly is 'increased orthogonalization'? This term is used twice in the manuscript, and seems to imply something specific but this is not made clear in the paper.

Reviewer #3 (Remarks to the Author):

Pu et al. present experiments and a computational model to resolve two seemingly contradictory effects of temporal order memory: 1) memory for the ordering of item pairs is improved when items have more temporal distance between them, presumably due to greater context dissimilarity; 2) memory for item pairs is improved when they are presented within the same event versus different events, presumably due to greater context similarity. Their model resolves this contradiction by assuming that temporal order memory is informed by an initial temporal context state. This context is reinstated at the start of each event, and is used to calculate similarity for recency judgments. They also examine contributions of local event position and event length. Through thoughtful (and thoughtfully depicted) experiments and controls, these results help to advance understanding of episodic memory, event segmentation and temporal perception. Each of these fields are well-developed in their own right, and forging connections between them is greater than the sum of their parts. The level of methodological and statistical detail would make replication straightforward. My concerns focus on clarifying the theoretical implications of this work, which most likely can be addressed with more explanation.

Most critically, I think the authors need to clarify the meaning and plausibility of what occurs at event boundaries in their model, as this contributes to the theoretical contribution of this manuscript. When the authors describe their resetting mechanism, they state that "event boundaries reset temporal context by recovering a certain proportion of the first contextual representation". To me, the 'resetting' mechanism seems a bit more like reinstatement of context the first item, rather than resetting or clearing memory. I suppose the reset could be to the first item's context, but usually when talking about retrieving and updating context based on a specific prior time, I would intuit this as reinstatement, as is done during the memory test. Perhaps this is meant to reflect a different type of representation, as the authors also "reasoned that there must be some abstract, schematic and generalizable information that is activated in the beginning and recovered at the event boundaries." This sounds qualitatively different than a temporal context state defined at the beginning of each list, although perhaps the idea is that such a state incorporates more abstract features, yet is simplified to take a similar structure to temporal context? That is, might it be intuited that really C1 should be a combination of a nontemporal, preexperimental, C0 along with temporal context C1? This is more than a matter of semantics or definitions, as it gets at the heart of the model's cognitive mechanism which resolves the temporal order memory contradictions. Further, it is important to be clear how this relates to prior studies addressing the cognitive mechanisms of event boundaries. Finally, if 'resetting' is equivalent to reinstatement of the first item's temporal context, how would this apply in everyday life? Might context be reset/reinstated to the first context in one's day? I realize most models make simplifying or idealized assumptions, yet I think the authors need to provide more explanation to justify the plausibility and intuition of this mechanism.

I also don't follow the intuition for how the proposed model can account for the local primacy effect. To attest to my point, in the description the model, there is no reference to this effect, or even the word 'primacy'. So if the authors could please make the connection more explicit, that would be ideal to relate their novel finding to their novel model.

Also, I'm sorry if I missed this, but how does the model make errors? Simply due to the noisiness of context, or is only a subset of context reinstated during test?

Further, I am less convinced of the complete novelty of the local primacy effect. Although the results have the added benefit of controlling for list position carefully and consider non-boundary items, I do think they need to make more contact with prior literature establishing improved memory (including temporal order memory) for event boundary items. Such a result would fall under the authors' "hypothesis that event boundaries cause an advantage in TOM for earlier local event positions". Some of the earliest segmentation work by Zacks (et al. 2007) and Radvansky (Pettijohn & Radvansky, 2012) also highlight improved memory at boundaries. (Clewett et al. 2019 reviews some of this literature in Section 6.) From my perspective, it seems that the model incorporates previous theoretical understanding---that event boundaries modulate representations and improve memory---and due to a slowly changing context state, this is carried into subsequent items, such that it is strongest at position 1 but carries over into other early positions? This still captures an important interaction and generalization of improved memory for boundary items, in particular the contribution of temporal context. Nonetheless they should credit previous theoretical and empirical work, and relate their developments to that work.

Response letter to reviewers

Re: Revision of NCOMMS-20-50772-T

Dear reviewers,

We would like to thank you for all the valuable and inspiring comments and suggestions, which have helped us to significantly improve the quality of our work. We have answered all of your questions and incorporated these suggestions within our manuscript. We summarized the reviewers' questions one by one, and our responses, in blue are detailed immediately after each question. Those changes are highlighted within the manuscript as well.

Again, we thank the reviewers for your time and close attention to our work.

Sincerely yours,

Yi Pu, Xiangzhen Kong, Charan Ranganath and Lucia Melloni

REVIEWER COMMENTS

Reviewer #1 (Remarks to the Author):

The paper provides strong evidence that event boundaries both reduce TOM across events and enhance TOM within events. The authors further show that this latter benefit may be due to a “local primacy effect” whereby items presented early on in an event are better remembered than items later in an event, even when testing across-event memory. Importantly, they propose a new model whereby boundaries are associated with a reset that recovers a “beginning” context that drifts slowly across the event. This set of findings and the proposed model reflect an important theoretical contribution towards understanding how event segmentation may improve memory, suggesting there are lingering benefits after an event boundary related to contextual resetting that fade over time. My major concerns, described below, are mainly related to some interpretation of the empirical data, to the possible high sensitivity of the model to event position, and to the consideration of the prior literature.

Re: We would like to thank the reviewer for the positive appreciation of our work and for the insightful comments provided, which have helped us to significantly improve the quality and focus of our work. We have addressed each of the questions one by one below.

Empirical data

Q1. In general, the authors do an excellent job controlling for possible confounds. However, there are a few places I am concerned about list position confounds in particular. In experiment 1, the use of a no boundary list is an excellent control, however boundary within and boundary across are still directly compared. Given that task diagram shows that the within event condition contains items from more primacy and recency positions in the list, I would recommend that comparison be removed. Alternatively, the authors could rerun the analysis excluding the first and last within event condition, which would presumably bias the effect in the reverse direction. Again, in experiment 2, the positions of the within versus across lag 1 appear biased to show the boundary effect on TOM. That is, the within lag 1 condition occupies more primacy and recency positions than both across-event conditions. Subsampling may be necessary to address this issue.

Re: We agree with the reviewer that the list positions were not perfectly matched for the “boundary effect” in Experiment 1&2. Several reasons motivated us to include the contrast between within-event and across-event pairs in Experiment 1. First, we aimed to replicate previous studies and to benchmark our results against prior work (e.g., Heusser et al., 2018). Second, we also aimed at evaluating the contribution of such a fact in our control condition, in which we took the same list positions and assigned those to matched within and matched across boundary conditions. In that control condition, we did not observe a statistical difference between matched within and across-event pairs, leading us to conclude that the boundary effect goes beyond a mere list position effect. We are nonetheless in agreement with the reviewer that list position might play a role in affecting TOM. Following the excellent suggestion, we added a control analysis in the revised manuscript in which the first and last item pairs were removed from the within-event pairs. The effect of better TOM for within vs. across-context pairs still

held for both Experiment 1 ($t(25) = 3.022, p = 0.0057$, two-tailed, 95% CI = 2.702 to 14.26) and Experiment 2 ($t(26) = 4.283, p = 0.0002$, two-tailed, 95% CI = 10.92 to 31.07) (p.6, p. 8).

Q2. Again in experiment 3, there is commendable effort put into matching the list positions of the within-event conditions that cleverly allows comparisons of early versus late event effects. However, the across event condition with shorter events/earlier event positions (3&3 for event 4 list vs 3&5 for event 6 list) is biased to show a short event/early event position enhancement because those also occupy more primacy and recency positions in the list (matching on average position doesn't help since primacy and recency counteract, even though they should both enhance memory). This is not a fatal flaw, as the next experiment matches the positions, but I think the authors should not report that effect in experiment 3 without significant qualifiers describing this confound (otherwise, again, subsampling could help).

Re: Thanks for the suggestion. In the revised manuscript we included a qualifier regarding a confound of list positions for across-event pairs in Experiment 3 to better motivate Experiment 4 (p.11). For this study, we opted against subsampling for two reasons. First, because too few trials (2trials/sequence * 7 sequences = 14trials) would have remained after subsampling, and second, because Experiment 4 addressed this confound directly.

Q3. In experiment 4, early versus late comparisons are perfectly matched, but the temporal distance effect pairs are not (by definition). The problem is, the longer pairs in grey include really early list items (the 2nd) which may be very easy to remember, while the shorter pairs do not contain any primacy or recency items within three positions from the start or end of the list. The authors should again subsample, dropping these first long-item pairs and/or qualify their description of the distance effect based on this potential confound.

Re: Since the two pair types cannot be perfectly matched even after subsampling (because longer pairs taking earlier items and shorter pairs taking later list positions), we opter for qualifying the effect by adding a description of such a confound in the revised manuscript as suggested (p.12).

Model

Q4. The proposed model is appealing in that it reconciles boundary and distance effects on TOM, but a have a couple concerns. First, it is somewhat bewildering that the model is so insensitive to parameter values. It would be important to show that at extreme values (e.g., very low reset,) the fit drops.

Re: Thanks for bringing this point up, which allows us to better explain our model. As suggested by the reviewer, in the revised manuscript (p.16 & Fig. S6), we now show two examples when the model could not explain the behavioral data ($p=0.01$ and $\lambda=0.05$; $p=0.02$ and $\lambda=0.06$). We did not mean to imply that our model is insensitive to parameter values. The original Fig. 6 only shows significant correlations between simulation results and behavioral results across all experiments for each set of parameter values. Such significant correlations however, do not imply that all the effects in each behavioral experiment have been correctly recovered by the model. In the revised manuscript, we added these necessary constraints (i.e., for d2-d1: within>across, within boundary > within no boundary, across boundary < across no boundary,

within Lag1>acrossLag3>acrossLag1, within early > within late, across early > across late, across longer lag > across shorter lag) and updated Fig. 6 (p.18). The area enclosed by red line in Fig. 6 includes the parameter values that yield model outputs which can recover all the effects in the four behavioral experiments. As shown in the updated Fig. 6, not all parameter values can produce model outputs that both correctly recover and significantly correlate with all the effects across experiments.

Q5. Critically, it seems that the within versus across effect would be very sensitive to the actual event positions sampled. If you are computing d_2-d_1 on position 2 versus 3, that will be much more difficult a judgment than 2 versus 5. So it seems like performance might be less sensitive to there being a restart at the boundary as it is to the exact positions sampled for testing. This would explain why the simulation recovers the across early versus late effect, since the positions sampled are closer in the late versus early condition (note, fig 5D is mislabeled “across early” in both). Additional simulations should be included to address these concerns (e.g., comparing within versus across while matching the absolute value of d_2-d_1).

Re: We thank the reviewer for these key questions. We apologize for the mislabeling of the figure (now corrected), and also that we were not clear about the evidence for the local primacy effect. It is not the case that the positions sampled are closer in the late than in the early condition. We now explicitly clarify that, in Experiment 4, the across early and across later event positions were matched for both list positions and the number of the intervening items. In this experiment, there is no case in which the positions sampled are closer in the across late than in the across earlier condition.

One potential source of confusion here is that the differences in serial positions of items within an “event” do not linearly correspond to the values of d_1 and d_2 in the model. In the model, d_1 and d_2 reflect distances of the items relative to the first one. The Memory Index (d_2-d_1) is larger for within versus across-event pairs in model simulation because of the “reset” process at event boundaries. Supporting evidence for this idea can be found in the simulation of Experiment 1 (Fig. 5A), where the boundary condition and the no boundary condition were directly compared while the list positions were matched. Simulation results show that only in the boundary condition, d_2-d_1 is larger for within-event pairs than across-event pairs, but not for matched positions in the no boundary condition (Fig. 5A). This excludes the possibility that the difference between within versus across-event pairs is simply due to different event/list positions sampled, and strongly argues for the importance of the reset process caused by event boundaries in affecting d_2-d_1 . In addition, we also subsampled the within-event pairs by removing the first pair in the simulation in Experiment 1, now leading to across-event items taking earlier list positions than the within-event pairs. Nevertheless, we still get larger d_2-d_1 for within versus across-event pairs (e.g., model outputs for within vs. across-event pairs is 0.0476 vs. 0.0114, when $p=0.02$, $\lambda=0.2$), suggesting that the reset process does play a significant role in affecting d_2-d_1 .

Q6: It would also be helpful to show whether and why the model predicts the local primacy effect given that the drift rate is constant.

Re: We thank the reviewer for the important question. The drift rate defines how much change occurs from one context to its immediately next context. Although it is constant, the change

between two neighbouring contexts relative to the first context is not. That is, the amount of Δd ($\Delta d = d_{t+1} - d_t$, where d denotes the distance of each context to the first context) across positions is not constant. The more dissimilar the contextual representation becomes to the first contextual representation, the smaller the amount of Δd will become. This is because compared to the contextual representation which is more similar to the first contextual representation, fewer proportion of the contextual representation that is more dissimilar to the first context can continue to become different from the first contextual representation.

We use an intuitive example to explain why Δd becomes smaller as the contextual representation becomes more dissimilar to the first contextual representation. For instance, if there are 1000 zeros in the first context vector, and the drift rate is 5%. In the second position, we would expect to get 950 zeros and 50 ones in the context vector. Thus, the change from the first context to the second one is 50. In the third position, although the drift rate is 5%, we would not expect to get 900 zeros and 100 ones. This is because while the 950 zeros in the second context vector can change to ones with the probability of 5%, the already existing 50 ones can also change back to zeros with the same probability of 5%. Therefore, in the third position, we would expect to get an average of $950 * 5\% + 50 * (1 - 5\%) = 95$ ones. Thus, the cumulated change of the third context relative to the first context is 95, and the change relative to the first contextual representation from the second to the third position (i.e., Δd between the second and the third position) is $95 - 50 = 45$, which is less than 50 (50 is the amount of Δd between the first and the second position). This example demonstrates why it is the case that the more dissimilar the contextual representation becomes to the first contextual representation, the smaller the amount of Δd becomes, although the drift rate is constant.

At event boundaries, participants recover a certain proportion of contextual information from the first context, such that contextual representations at event boundaries will become more similar to the first contextual representation compared to contextual representations of pre-boundary items. This therefore, will lead to increased amount of Δd at event boundaries. As a result, the **accumulated Δd** between two items is larger for positions closer to an event boundary than farther away from an event boundary within an event (i.e., earlier versus later event positions). Since the accumulated Δd between two items is the distance of the two items relative to the first context (i.e., $d_2 - d_1$ in Figure 4D). E.g., the accumulated Δd between item $t+2$ and item t is $d_{t+2} - d_t$, which is equal to $(d_{t+2} - d_{t+1}) + (d_{t+1} - d_t) = d_{t+2} - d_t$, where d is the distance of the item's context relative to the first context). This explains why the present model can account for the local primacy effect caused by event boundaries.

We added the explanation in the revised manuscript (pp. 21-22).

Q7. Although the model explains TOM, it cannot account for the prominent related effect that boundaries increase perceived temporal distance (e.g., as shown in the Ezzyat and Rouhani references as well as others). Indeed, it seems like it would make the opposite prediction if using the $d_2 - d_1$ metric. If the model can account for it, the authors should include relevant simulations, as that would be a significant contribution. Otherwise, some discussion of this limitation should be included as it is a highly relevant effect in the literature.

Re: This is an excellent point. When reviewing the papers suggested by the reviewer and others, we found mixed results on how boundaries affect perceived temporal distance/duration of two items. Indeed, Ezzyat and Davachi (2014) reported that event boundaries increase perceived temporal distance. However, in one experiment reported by Rouhani et al. (2020), they did not observe that event boundaries or the presentation distance affect perceived temporal distance. In a replication experiment, they found that event boundaries significantly increase perceived temporal distance only for neighboring items. The effect was marginal for items separated by one item, and was not even significant for items separated by three intervening items (Fig 3D in Rouhani et al. (2020)).

Another study by Bangert et al. (2020), however, found that event boundaries decrease perceived temporal duration. Participants were required to compare the temporal duration of a movie clip to a 5s' reference interval, and to judge whether the movie clip is longer or shorter than the reference interval. It was found that movie clips containing an event boundary were more likely to be judged as shorter than the reference interval, while movie clips containing no event boundary were more likely to be judged as longer than the reference interval, although the lengths of the two types of movie clips were identical.

The results so far point to a complex scenario of how event boundaries affect temporal distance/duration estimation. More studies are needed to fully and rigorously examine such an effect.

Another important and related question concerns whether the decision-making process is identical for temporal distance/duration judgments and temporal order judgments. This is key to the question of what metric should be developed in a model as an appropriate proxy for each of the two effects (we think the encoding process should be identical regardless of the test though). In thinking about this question, we realize that a reference is necessary to make recency judgments, however this might not necessarily be the case when making temporal distance/duration judgments between two items. For the latter, the decision can alternatively be based on the similarity of the two items (although this operation could not allow for recency judgments).

- If we assume that a reference is used for judging perceived temporal distance, we can still use $d_2 - d_1$ as the proxy for temporal distance estimate. Under those assumptions, the model would predict that the perceived distance is longer for within-event pairs than across-event pairs, and longer for pairs separated by more intervening items than items separated by fewer intervening items (see Fig. 5B in the manuscript as an example).
- Yet, if we assume that the judgment of temporal distance/duration is based on comparing the similarity of the two probed items, whereby the more dissimilar the contextual representations between two items are, the farther apart the two items would be perceived. Under those assumptions, the model would predict that the perceived distance is shorter for within-event pairs than across-event pairs, and longer for pairs separated by more intervening items than pairs separated by fewer intervening items (see one example of

simulation below when $p=0.02$, $\lambda=0.2$, MI: Within Lag1 > Across Lag1 > Across lag3).

Since the existing empirical results are mixed, both predictions can find empirical support (e.g., Bangert et al., 2020; Rouhani et al., 2020). As mentioned above, further studies are needed to first fully and rigorously examine how event boundaries and temporal distance affect perceived temporal distance. Once the phenomenon is well established, the question of which metric is an appropriate proxy for temporal distance judgments could be tackled. This is because the decision-making process between recency judgments and temporal order judgments might significantly differ from each other, although the two cognitive tasks share superficial similarity. In the revised manuscript, we added a brief discussion on this point as suggested (pp. 23-24).

Literature

Q8. When citing a relatively new finding like the boundary effect on TOM, the first paper(s) to show it should be referenced. The first, as far as I know, was DuBrow & Davachi (2013). This paper was cited in the methods for the design but not for its theoretical contribution. Moreover, the paper reports both a boundary effect on TOM (reduced) and a distance effect on TOM (enhanced) that the current manuscript suggests had not previously been demonstrated to “co-exist in the same experiment.”

Re: We thank the reviewer for this observation. Following the reviewer’s suggestion, in the revised manuscript we cite DuBrow & Davachi (2013) (reference [7] in the revised manuscript) when reporting the boundary effect (p. 3) and removed the statement “...had not previously been demonstrated to co-exist in the same experiment”.

Q9. The current manuscript cites Jenkins and Ranganath (2016) for the “temporal distance effect.” That paper only probes a single temporal distance so it is impossible to show the distance effect. The authors should reference papers that do show the effect, e.g., originally Ynetma & Trask (1963). For a more recent demonstration, see St Jacques et al (2008). They should also provide references for this statement “hippocampal representations are more dissimilar for experiences further apart than experiences close in time” and consider drawing on the animal literature.

Re: Corrected (pp. 3-4). We added the papers mentioned by the reviewer, and also added animal studies and other human studies as suggested in the revised manuscript.

Q10. In the next paragraph, Ezzyat and Davachi (2014) is somewhat misrepresented, as essentially the same number of regions showed within-event similarity as across-event similarity. Later in the manuscript (pg 14) it is referenced along with the Jenkins paper and another preprint as showing more similarity between item pairs with shorter vs longer lags, but neither

of the peer-reviewed papers show that effect. Indeed, they don't even manipulate lag, and, even if they did, that would be meaningless due to the slow drift of the BOLD signal. Later again, in the discussion, the Ezzyat paper is referenced as showing TOM, which it did not test.

Re: We removed the reference on p.14 and also replaced Ezzyat and Davachi (2014) with DuBrow & Davachi (2013) as the reference showing TOM.

Q11. The authors reference the models in Horner et al (2016) and Rouhani et al (2020) as accounting for “either the ‘temporal distance’ or the ‘boundary effect’” but I don’t see anything in either model regarding the temporal distance effect.

Re: Corrected (p.13). In the revised manuscript, we state ‘*Previous models (e.g., Horner et al., 2016; Rouhani et al., 2020) have been put forward to account for the “boundary effect”, yet none, to our knowledge, aim to simultaneously account for both the “boundary effect” and the “temporal distance”, nor for the “local primacy effect”.*

Q12. At least two additional relevant models should be referenced: Howard et al (2015, Psych Review) which explicitly modeled TOM. Henson (1998, Cog Psych) which represents items in terms of their distance from the starting point in a conceptually similar way to present model’s recency judgment metric.

Re: Added as suggested (p. 29).

Q13. A quick submission of the reference list to this field-specific gender citation balance tool (<https://postlab.psych.wisc.edu/gcbialyzer/>) showed a strong bias in favor of male lead authors. I would strongly recommend expanding the references to include more contributions from the many women authors in this subfield.

Re: This is an excellent point. We have carefully reviewed the manuscript noting the gender of the authors. In doing so, we included as many female lead authors as possible. To our knowledge, the main female leading authors examining event segmentation and cognition are Dubrow (from Davachi’s lab, and now a PI in her own), Rouhani, Shin. We agree with the reviewer that there is indeed a bias in our reference list towards male leading authors. This might reflect the fact that probably more male leading authors have published papers in this field. Such a status quo is also reflected in the papers recommended by the reviewers, whose leading authors are mainly males (e.g., in Q12). We fully embrace the idea that gender balance is something to strive for in research, and support initiatives such as the one from Brad Postle Lab. If the reviewer has specific suggestions in this regard, we would be happy to include them in our research.

Reviewer #2 (Remarks to the Author):

Pu et al present four behavioral experiments and a computational model addressing the role of event boundaries on temporal order memory (TOM). The authors present subjects with a series of objects temporally separated by color event boundaries, and manipulate the number of events and inter-item lag for TOM judgments. The authors show that boundaries increase TOM within events and decrease TOM across events. They then show that manipulating event length leads to local primacy and demonstrate a boundary effect and temporal distance effect in the same paradigm. The data can be explained by a novel temporal context style model in which the first list context is reinstated at every event boundary.

There are several interesting findings here and the authors have conducted a nice series of well-controlled experiments. My main concerns relate to the model, which is a significant part of the novelty and impact of the manuscript, but has shortcomings that need to be addressed.

Re: We are grateful for the reviewer's comments and enthusiastic evaluation of our work. We hope that the revision has addressed the questions raised by the reviewer.

Major Comments

Q1. The authors create a novel computational model that explains the boundary effect and the temporal distance effect, with the main innovation being to have the first context reinstated at each event boundary. My first comment is about this mechanism/assumption: is there any prior behavioral or neural evidence that this is actually what people do at event boundaries that would motivate including it as a mechanism in the model, other than the fits to their own data? As a potential explanation for organization in memory, it seems much more plausible and general that context would simply shift more at event boundaries (Horner et al 2016; DuBrow et al 2017) than be reinstated from an arbitrarily defined timepoint at the start of a list.

Re: We thank the reviewer (along with Reviewer 3) for raising the important question on the rationale of our model. We agree that the rationale for the model could be improved, and in our revised manuscript, we have made it clearer.

In brief, as Reviewer 3 suggested, it is reasonable to think that participants reinstated the context of the beginning of the experiment. Our model fits with existing theories of event cognition (e.g., Zacks et al., 2007) which suggests that participants must generate a new event model following an event boundary. Forming an event model is not simply random, but rather, it is an attempt to make sense of the information that is being processed (Franklin et al., 2020; Shin & DuBrow, 2020). In complex, real life events, we often rely on memory retrieval when building a new event model (Cohn-Sheehy et al., 2020; Franklin et al., 2020; Lu et al., 2020; Shin & DuBrow, 2020). For instance, when crossing a doorway en route to a particular location, it makes sense to reinstate information from the beginning of your journey, in order to recall what made you leave the room in the first place.

In this and other temporal order memory experiments, events are much simpler, but it is nonetheless sensible to reinstate information from the beginning of the experiment, when

participants first learned about the structure of the task (i.e., changing color frames, sequential structure, etc.), as well as any other schematic information which may facilitate learning of the present new event. Although it is conceivable that context shifted randomly at each event boundary, it seems unlikely, as this would imply that the participant would be quite confused at each context shift as opposed to recognizing it as a predictable aspect of the task structure (Shin & DuBrow, 2020).

We implemented such an idea of non-random change (i.e., systematic change) at event boundaries by assuming first that a certain proportion of the first contextual information is reinstated at each event boundary, and second that the first contextual information contains both pre-experimental contextual representation (e.g., the task structure, schema) and the first contextual representation. We have clarified these issues in the revised manuscript (pp.13-14).

Beyond the point that it is reasonable to assume that context shifts at boundaries are not entirely random, it is also important to note that a random context shift model cannot account for the results. When we performed simulations (p.18 & Fig. S4, S7, S8) based on the idea that event boundaries cause a random sharp change, the results could not account for both the “boundary effect” and the “temporal distance effect”. By incorporating such a reinstatement process at event boundaries, our model can successfully account for both effects.

Q2. Related to the previous point, the model fits the data across a wide range of parameter values, but the values that are highlighted $p = 0.02$ and $\lambda = 0.2$. This tells us that the degree to which context moves from moment to moment is ten times smaller than the degree to which context at every event boundary moves back towards the context that existed for the first list item. This seems unlikely to be true in reality, but if the authors disagree they should motivate this result more fully. I also think it would be helpful to present fits to the data (Figure 5) using other parameter values.

Re: As noted by the reviewer, we started with the parameter set of $p=0.02$ and $\lambda=0.2$ in our model simulation. Our point here was to follow an existing model (Horner et al., 2016) as a starting point. Horner et al. (2016)’s model assumes that the shift rate is *eight times* larger than the drift rate. Therefore, for simplicity, we first tried the parameter values in which the reset rate is ten times larger than the drift rate. This is because our common assumption is that the shift rate should be larger than the drift rate.

In the following analysis, we show in Fig. 6 (Fig. 6 was updated in the revised manuscript to answer Q4 raised by Reviewer 1) that there are also other sets of parameter values, in which reset rate is not ten times larger than drift rate. These parameter values can also recover all the effects shown in the behavioral experiments and significantly correlate with all behavioral results (see Fig. S5 as an example, when $p = 0.1$, $\lambda=0.15$).

Q3. The authors find in Exp 2 that Within Lag1 performance is better than Across Lag 1 performance and, more surprisingly, Across Lag 3 performance. This is one of the key findings that motivates their argument against a direct associative chaining or similarity mechanism. However, the authors do not consider the potential for some kind of strength based process (e.g. Jenkins & Ranganath 2010) to be at play. The task instructed participants to directly link items

in the same event, which could have led to weakened associations between the last item of one event and the boundary item of the next event. This would have produced the Within Lag1 and Across Lag 1 difference that they found. Introducing an additional item strength mechanism for the longer lags could potentially support the intermediate performance in the Across Lag 3 condition. The authors should compare an alternative model that includes such a mechanism, for which there is prior data as motivation.

Re: This is an important question. We decided not to include an item strength based mechanism (ISBM) in the model, because evidence from previous behavioral work investigating the ISBM in recency judgments often shows mixed results (see also Howard et al., 2015 for a review), and because the predictions based on the ISBM are in contrast with empirical observations (Brown et al., 2000). We detailed the reasons below.

First, evidence supporting an item strength-based mechanism (ISBM) in recency judgments is not conclusive. In fact, previous experiments testing the hypothesis that the discrimination of recency is based on a comparison of the strengths of memory traces often yield mixed results. By using item repetition as a way to strengthen item memory (since repetition can generally improve recall and recognition memory experiments), Peterson et al. (1967) found that two presentations of an item do not lead to the last occurrence being judged more recent than if only a single occurrence was judged, suggesting no effect of repetition. Later, Peterson et al. (1969) expanded the experiment and found that while spaced repetitions do not change the recency judgments compared to single presentation, massed repetitions do lead to the judgment for the last occurrence of the item being more recent. The data was then interpreted as supporting the ISBM, because massed repetition is assumed to strengthen a single memory trace, while spaced repetition is assumed to produce multiple memory traces, and the item with a stronger memory trace would be judged more recent. However, findings from a later study (Morton, 1968) argue against such an idea by showing that participants tend to choose the repeated item (e.g., AAB, ABB) as either the more recent or earlier item depending on the task requirement. This result suggests that choosing a repeated item may simply reflect strategic guessing (Hintzman, 2010). That is, if there are three events A1, A2, B, and they could occur in any order. In four out of six orders, A is first and in four of the six orders, A is last. Without reliable recency information, the most rational choice in either task (which one was presented later or earlier) is to choose the repeated one.

Second, predictions based on the ISBM are in contrast with empirical observations. The reviewer kindly suggested that the reason why TOM is better for Across Lag3 (items 3 vs. 3') than Across Lag1 (items 3 vs. 1') and worse than Across Lag1 (items 1 vs. 3) might be attributed to the ISBM. However, if people choose the item with stronger memory strength (i.e., better item memory) as the more recent item, we would predict better TOM for Across Lag1 than both Across Lag3 and within Lag1 in our experiment. This is because in Across Lag1 (i.e., items 3 vs. 1), the boundary items (i.e. item 1 in an event), which have been repeatedly shown in previous papers to be remembered better (see Clewett et al., 2019 for a review), are included in the probed pairs and happen to be the more recent item as well. Therefore, the accuracy for recency judgements should be very high for Across Lag 1. In contrast, no boundary items are included in both Across Lag3 and within Lag1 pairs. Thus, the accuracy for recency judgments should be

lower for both probed pairs than Across Lag1. These predictions are apparently in contrast with our empirical observations.

We also would like to clarify the experimental instructions: participants were required to link neighboring items together regardless of the color change (p.25), not just to link two neighboring items within events.

The reviewer mentioned fMRI data from Jenkins and Ranganath (2010) as supporting evidence for the ISBM. We assume that the reviewer might have meant Jenkins and Ranganath (2016), since no evidence was found to support the ISBM in Jenkins and Ranganath (2010). Jenkins and Ranganath (2016) found that activation in perirhinal cortex during encoding of objects “predicted whether an object would be *judged* more recent at test”. This is a critical point because perirhinal activation was predictive of subjective recency *regardless of accuracy*. In contrast, hippocampal activation was predictive of accurate recency judgments, and this effect was interpreted as relevant to temporal context encoding. Thus, the key point made by Jenkins and Ranganath (2016) is that item strength might be a *heuristic* used by participants to make recency judgments, not that it is a mechanism that necessarily supports accurate judgments. Strength differences can potentially support recency discriminations when the temporal distances between probed items are large. With the short temporal distances between probe items used in this and other recent TOM studies, a strength-based heuristic would be especially unreliable. The present paradigm (like that of DuBrow & Davachi, 2013 and Rouhani et al., 2020) was not intended to test the contributions of item strength to TOM, it would be of interest for future studies to do so with longer lags.

We fully agree with the idea that temporal order memory might be supported by multiple mechanisms, under different circumstances. However, our model, like other previous TOM models (e.g., Brown et al., 2000; Horner et al., 2016; Howard et al., 2015) was intended to provide the most parsimonious account for the present results. In doing so, our present account resolves a long-standing paradox in the memory literature on TOM.

The above points are discussed in the revised manuscript (p.23, lines 586-593).

Q4. The authors report collecting confidence judgments on the TOM judgment but then do not discuss them. Were the boundary and lag conditions matched in terms of confidence? Knowing this would help answer the question of whether another mechanism, such as a strength process or associative reinstatement, is influencing recency judgments.

Re: We followed the reviewer’s suggestion and performed statistical analyses on confidence ratings. No significant difference was found between boundary condition and no boundary condition (Experiment 1), and no significant difference was found between Across Lag 3 and Across Lag1 (Experiment 2). The full set of results are reported in the supplementary materials Fig. S2 in the revised manuscript.

In our view, confidence ratings on item recognition increase continuously with item memory strength, but confidence ratings on recency judgments, as adopted in our task would not be expected to provide much information about item memory strength. Nevertheless, the addition of

the confidence rating results helps to round out the presentation in the revised manuscript, and it may inspire readers to test new hypotheses in future studies.

Q5. The authors don't account for previous data showing evidence for reinstatement of direct associations during recency judgments (e.g. DuBrow & Davachi 2014), only mentioning in the Discussion that they did not observe evidence of such a process in their data. While they do show response times in Figure S1 sorted by the position of the first item in a pair, what they need to show is RTs as a function of inter-item distance and boundary condition (i.e. RTs to match the analyses in Figures 1, 2, and 3). However, even if they do not find evidence for inter-item reactivation in response times, they should still explain a more plausible explanation for why their data differ from prior work.

Re: The original Fig. S1(now Fig. S3) aimed at demonstrating that there was no evidence supporting backward or forward scanning during recency judgments in our task. As suggested, we added the results on reaction time (RT) in the supplementary materials Fig. S1 to parallel Fig. 1-3. To examine whether there is evidence supporting reinstatement of direct associations, we compared RT in Across Lag3 and Across Lag1 in Experiment 2. We reasoned that, if intervening items are sequentially reinstated during recency judgments, we should expect longer RT for Across Lag3 than Across Lag1. However, no significant difference was found (see Fig. S1). This pattern of results parallels the behavioral findings from DuBrow and Davachi (2014), where RT was not found to be significantly different between no switch long lag and no switch shorter lag. We also carefully reviewed the fMRI results and results from a follow-up behavioral experiment in DuBrow and Davachi (2014). We found that the evidence supporting reinstatement of intervening sequences seems to be mainly limited in no switch condition. Thus, since our behavioral data did not provide evidence supporting item reinstatement and previous data testing this idea seem inconclusive, we opted not to include such a mechanism in the model. In the revised manuscript, we have included an explanation of why such a sequential reactivation of intervening items mechanism was not included in the model as suggested (pp.29-30).

Q6. There are many places where sentences and paragraphs are more complicated and/or less clear than they need to be, and this definitely affects the impact of the paper. Generally, I try not to comment in reviews on elements of style, but in this case I really think the writing is getting in the way of the message. For example, in the Introduction, the authors state: "Thus, the basis aspects of how events boundaries structure temporal memories remain to be understood. For instance, whether boundaries enhance and/or impair TOM for items within and across events, respectively...". The first time I read this I was confused because it seems to suggest that prior work has not examined within and across event temporal order memory, which is not the case (and is not actually what the authors mean). The fact that the authors will investigate a novel comparison of a boundary condition with a no-boundary baseline (Exp 1) is not as clear as it needs to be--this is an important aspect of the paper that is confused by the construction of these sentences and the paragraph they are in. This problem exists throughout the manuscript--I think the authors should undertake a thorough revision of the writing, in order to make the findings and their impact clearer.

Re: We thank the reviewer for this comment. For the revision, we proofread and edited the manuscript, hoping that the messages are conveyed in a more straightforward manner.

Minor Comments

Q7. What exactly is 'increased orthogonalization'? This term is used twice in the manuscript, and seems to imply something specific but this is not made clear in the paper.

Re: Corrected (p.19, p.21). We meant “increased difference between the contextual representations of the probed items relative to the first contextual representation (i.e., increased magnitude of d_2-d_1 , Fig. 4D)”.

Reviewer #3 (Remarks to the Author):

Pu et al. present experiments and a computational model to resolve two seemingly contradictory effects of temporal order memory: 1) memory for the ordering of item pairs is improved when items have more temporal distance between them, presumably due to greater context dissimilarity; 2) memory for item pairs is improved when they are presented within the same event versus different events, presumably due to greater context similarity. Their model resolves this contradiction by assuming that temporal order memory is informed by an initial temporal context state. This context is reinstated at the start of each event, and is used to calculate similarity for recency judgments. They also examine contributions of local event position and event length. Through thoughtful (and thoughtfully depicted) experiments and controls, these results help to advance understanding of episodic memory, event segmentation and temporal perception. Each of these fields are well-developed in their own right, and forging connections between them is greater than the sum of their parts. The level of methodological and statistical detail would make replication straightforward. My concerns focus on clarifying the theoretical implications of this work, which most likely can be addressed with more explanation.

Re: We thank the reviewer for the constructive reviews, which were of great help when revising the manuscript. We addressed the questions raised by the reviewer and believe that those suggestions have helped us to significantly improve the revised manuscript.

Q1. Most critically, I think the authors need to clarify the meaning and plausibility of what occurs at event boundaries in their model, as this contributes to the theoretical contribution of this manuscript. When the authors describe their resetting mechanism, they state that "event boundaries reset temporal context by recovering a certain proportion of the first contextual representation". To me, the 'resetting' mechanism seems a bit more like reinstatement of context the first item, rather than resetting or clearing memory. I suppose the reset could be to the first item's context, but usually when talking about retrieving and updating context based on a specific prior time, I would intuit this as reinstatement, as is done during the memory test. Perhaps this is meant to reflect a different type of representation, as the authors also "reasoned that there must be some abstract, schematic and generalizable information that is activated in the beginning and recovered at the event boundaries." This sounds qualitatively different than a temporal context state defined at the beginning of each list, although perhaps the idea is that such a state incorporates more abstract features, yet is simplified to take a similar structure to temporal context? That is, might it be intuited that really C1 should be a combination of a nontemporal, preexperimental, C0 along with temporal context C1? This is more than a matter of semantics or definitions, as it gets at the heart of the model's cognitive mechanism which resolves the temporal order memory contradictions. Further, it is important to be clear how this relates to prior studies addressing the cognitive mechanisms of event boundaries.

Re: We would like to thank the reviewer for bringing up this excellent point. Our previous manuscript focused primarily on the computational elements of the model, and we agree that it is important to clarify how the model relates to the processes that may be plausibly engaged in this experiment. We have revised our manuscript to make these points clearer (pp.14-15). We concur with reviewer 3's intuition, that participants are recalling information from the first context,

which contains both nontemporal, preexperimental context along with the first temporal context. In our revised manuscript, we state:

“Since recent neuroimaging and psychophysics studies have shown that event boundaries can cause a sharp change (i.e., shift) in brain/mental state (e.g., Baldassano et al., 2017; Clewett et al., 2020), it has been hypothesized that such a sharp shift is caused by a faster random change in temporal context at event boundary (e.g., DuBrow & Davachi, 2017). Our model, however, assumes that the shift in temporal context is not a random process, but instead is based on reinstatement of the pre-experimental context.

This assumption fits with existing theories of event cognition (e.g., Zacks et al., 2007), which postulate that following an event boundary participants generate a new event model. Forming an event model is not simply random, instead it is an attempt to make sense of the information that is being processed (Franklin et al., 2020; Shin & DuBrow, 2020). In complex, real life events, building a new event model often relies on memory retrieval (Cohn-Sheehy et al., 2020; Franklin et al., 2020; Lu et al., 2020; Shin & DuBrow, 2020). For instance, when crossing a doorway en route to a particular location, it makes sense to reinstate information from the beginning of the journey, in order to recall what made us leave the room in the first place. If mental context randomly shifted at each event boundary, one would possibly forget one’s destination after crossing the first doorway. In our and other temporal order memory experiments, events are much simpler, but it is nonetheless sensible to reinstate information from the beginning of the experiment, when participants first learned about the structure of the task (i.e., changing colour frames, sequential structure, etc.), as well as any other schematic information which may facilitate learning of the sequence.

In our model, we operationalized such a systematic change at event boundaries by assuming that a certain proportion of the initial contextual information is reinstated at each event boundary and that the initial contextual information contains both pre-experimental contextual representation (e.g., the task structure) and the first contextual representation of the list.” (pp.13-14)

We believe that the paper has been improved immensely by incorporating ideas about potentially plausible cognitive mechanisms engaged at event boundaries, and we thank the reviewer for these constructive comments.

In the future, more work with appropriate experimental designs should be done to further examine what information is activated at the beginning and reactivated at event boundaries. We have made some speculations in the Discussion (p.22).

Q2: Finally, if 'resetting' is equivalent to reinstatement of the first item's temporal context, how would this apply in everyday life? Might context be reset/reinstated to the first context in one's day? I realize most models make simplifying or idealized assumptions, yet I think the authors need to provide more explanation to justify the plausibility and intuition of this mechanism.

Re: This is another great suggestion. In our revision, we have added real-life examples, in which we consider the implications of random shifts vs. “resetting” context to a strategically important reference point. In one of the examples, we have now included in the text (see response to Q1 of

Reviewer 3 above), we consider navigating across several rooms in order to achieve a goal. A large and completely random shift in context following transitions between rooms would lead you to rapidly lose information about your goal and, for that matter, your destination. Alternatively, it is reasonable to think that, at each boundary, we reinstate information from the beginning of a journey in order to prevent important information from getting lost, which would assist us to achieve our goal. Of course, we do sometimes forget why we went to a particular room (Radvansky & Copeland, 2006), but this effect is much smaller than we would expect if our contextual state dramatically shifted in a random fashion at an event boundary.

Q3. I also don't follow the intuition for how the proposed model can account for the local primacy effect. To attest to my point, in the description the model, there is no reference to this effect, or even the word 'primacy'. So if the authors could please make the connection more explicit, that would be ideal to relate their novel finding to their novel model.

Re: We thank the reviewer for the important question. Please refer to the response to Q6 of Reviewer 1.

Q4. Also, I'm sorry if I missed this, but how does the model make errors? Simply due to the noisiness of context, or is only a subset of context reinstated during test?

Re: By asking how the model makes errors, we assume that the reviewer might ask the source of the uncertainty of the model outputs (i.e., what makes the model outputs not identical for each simulation). If so, the uncertainty of the model outputs is due to the random noise of the temporal context during encoding as the reviewer understood. We did not plot the error bar in the box plot of the simulation results, since the magnitude of the error bar is inversely proportional to the number of simulations. So the error bar is not informative in this case.

For the purposes of simplicity, in our model, we assume that the context signal of the probe items during encoding would be fully reinstated during retrieval. So the uncertainty of the model is just due to the random noise of the temporal context.

Q5. Further, I am less convinced of the complete novelty of the local primacy effect. Although the results have the added benefit of controlling for list position carefully and consider non-boundary items, I do think they need to make more contact with prior literature establishing improved memory (including temporal order memory) for event boundary items. Such a result would fall under the authors' "hypothesis that event boundaries cause an advantage in TOM for earlier local event positions". Some of the earliest segmentation work by Zacks (et al. 2007) and Radvansky (Pettijohn & Radvansky, 2012) also highlight improved memory at boundaries. (Clewett et al. 2019 reviews some of this literature in Section 6.) From my perspective, it seems that the model incorporates previous theoretical understanding---that event boundaries modulate representations and improve memory---and due to a slowly changing context state, this is carried into subsequent items, such that it is strongest at position 1 but carries over into other early positions? This still captures an important interaction and generalization of improved memory for boundary items, in particular the contribution of temporal context. Nonetheless they should credit previous theoretical and empirical work, and relate their developments to that work.

Re: This is a great point, which allows us to clarify how our data is linked to but also differ from previous findings. Indeed, as the reviewer pointed out, previous studies such as those mentioned by the reviewer have reported improved memory at event boundaries, which is in line with the general hypothesis that event boundaries improve memory. Nevertheless, we think that the local primacy effect differs from those effects and is novel. The reasons are two folded.

First, the local primacy effect reported in the present study describes a phenomenon in which memory improvement is strongest at the beginning of an event and gradually decreases as event positions move away from the event boundary. In contrast, the memory improvement reported by previous studies, is either only restricted to event boundaries or shows no gradual change as a function of the proximity to an event boundary (e.g., Gold et al., 2017; Heusser et al., 2018; Sonne et al., 2017; Swallow et al., 2009).

Second, the memory improvement reported in previous studies is in the domain of non-temporal memory, such as an increase in item memory or source memory (e.g., Heusser et al., 2018; Pettijohn et al., 2016), whereas the local primacy effect reported in the present manuscript is on temporal order memory. In the revised manuscript, we followed the reviewer's suggestion and added a discussion on how the local primacy effect is linked to but also differ from previous studies in the revised manuscript (p.20).

As for the model, it is true that previous theories mentioned by the reviewer propose improved memory for items in close proximity to event boundaries. However, as discussed in the manuscript, previous theoretical models such as the Event Horizon Model (Radvansky & Zacks, 2017) attribute the memory improvement accompanied by more event boundaries to decreased competition among items in the same event. This explanation cannot account for the local primacy effect, because it would predict *worse* TOM for pairs 2-4 in the "Event 6" condition relative to pairs 2-4 in the Event 4 condition in Experiment 3 (i.e., because the interference level is higher in the "Event 6" condition compared to the "Event 4" condition). This prediction is in contrast with our empirical observations. Our model differs from those theoretical models by introducing the slowing changing temporal context signal as pointed out by the reviewer, such that the benefit on TOM is stronger for positions closer to event boundaries than those farther away from event boundaries (see also the response to Q6 of Reviewer 1 for why the model can explain the local primacy effect). We included a discussion (p.20, p.23) on how our data is linked to the Event Horizon Model and added an explanation of how our model can account for the local primacy effect (pp. 21-22) as suggested.

References

- Baldassano, C., Chen, J., Zadbood, A., Pillow, J. W., Hasson, U., & Norman, K. A. (2017). Discovering Event Structure in Continuous Narrative Perception and Memory. *Neuron*, *95*(3), 709-721 e705. doi:10.1016/j.neuron.2017.06.041
- Bangert, A. S., Kurby, C. A., Hughes, A. S., & Carrasco, O. (2020). Crossing event boundaries changes prospective perceptions of temporal length and proximity. *Atten Percept Psychophys*, *82*(3), 1459-1472. doi:10.3758/s13414-019-01829-x
- Brown, G. D., Preece, T., & Hulme, C. (2000). Oscillator-based memory for serial order. *Psychol Rev*, *107*(1), 127-181. doi:10.1037/0033-295x.107.1.127

- Clewett, D., DuBrow, S., & Davachi, L. (2019). Transcending time in the brain: How event memories are constructed from experience. *Hippocampus*, *29*(3), 162-183. doi:10.1002/hipo.23074
- Clewett, D., Gasser, C., & Davachi, L. (2020). Pupil-linked arousal signals track the temporal organization of events in memory. *Nat Commun*, *11*(1), 4007. doi:10.1038/s41467-020-17851-9
- Cohn-Sheehy, B. I., Delarazan, A. I., Crivelli-Decker, J. E., Reagh, Z. M., Mundada, N. S., Yonelinas, A. P., . . . Ranganath, C. (2020). Narratives bridge the divide between distant events in episodic memory. *bioRxiv*.
- DuBrow, S., & Davachi, L. (2013). The influence of context boundaries on memory for the sequential order of events. *J Exp Psychol Gen*, *142*(4), 1277-1286. doi:10.1037/a0034024
- DuBrow, S., & Davachi, L. (2014). Temporal memory is shaped by encoding stability and intervening item reactivation. *J Neurosci*, *34*(42), 13998-14005. doi:10.1523/JNEUROSCI.2535-14.2014
- DuBrow, S., & Davachi, L. (2017). Commentary: Distinct neural mechanisms for remembering when an event occurred. *Frontiers in psychology*, *8*, 189. doi:ARTN 189 10.3389/fpsyg.2017.00189
- Ezzyat, Y., & Davachi, L. (2014). Similarity breeds proximity: pattern similarity within and across contexts is related to later mnemonic judgments of temporal proximity. *Neuron*, *81*(5), 1179-1189. doi:10.1016/j.neuron.2014.01.042
- Franklin, N. T., Norman, K. A., Ranganath, C., Zacks, J. M., & Gershman, S. J. (2020). Structured Event Memory: A neuro-symbolic model of event cognition. *Psychol Rev*, *127*(3), 327-361. doi:10.1037/rev0000177
- Gold, D. A., Zacks, J. M., & Flores, S. (2017). Effects of cues to event segmentation on subsequent memory. *Cogn Res Princ Implic*, *2*(1), 1. doi:10.1186/s41235-016-0043-2
- Heusser, A. C., Ezzyat, Y., Shiff, I., & Davachi, L. (2018). Perceptual boundaries cause mnemonic trade-offs between local boundary processing and across-trial associative binding. *J Exp Psychol Learn Mem Cogn*, *44*(7), 1075-1090. doi:10.1037/xlm0000503
- Hintzman, D. L. (2010). How does repetition affect memory? Evidence from judgments of recency. *Mem Cognit*, *38*(1), 102-115. doi:10.3758/MC.38.1.102
- Horner, A. J., Bisby, J. A., Wang, A., Bogus, K., & Burgess, N. (2016). The role of spatial boundaries in shaping long-term event representations. *Cognition*, *154*, 151-164. doi:10.1016/j.cognition.2016.05.013
- Howard, M. W., Shankar, K. H., Aue, W. R., & Criss, A. H. (2015). A distributed representation of internal time. *Psychol Rev*, *122*(1), 24-53. doi:10.1037/a0037840
- Jenkins, L. J., & Ranganath, C. (2010). Prefrontal and medial temporal lobe activity at encoding predicts temporal context memory. *J Neurosci*, *30*(46), 15558-15565. doi:10.1523/JNEUROSCI.1337-10.2010
- Jenkins, L. J., & Ranganath, C. (2016). Distinct neural mechanisms for remembering when an event occurred. *Hippocampus*, *26*(5), 554-559. doi:10.1002/hipo.22571
- Lu, Q., Hasson, U., & Norman, K. A. (2020). Learning to use episodic memory for event prediction. *bioRxiv*.
- Morton, J. (1968). Repeated Items and Decay in Memory. *Psychonomic Science*, *10*(6), 219-&. doi:10.3758/bf03331489

- Peterson, L. R., Johnson, S. T., & Coatney, R. (1967). The effect of repeated occurrences on judgments of recency. *J Verbal Learning Verbal Behav*, *8*, 591-596.
- Peterson, L. R., Johnson, S. T., & R., C. (1969). The Effect of Repeated Occurrences on Judgments of Recency. *J Verbal Learning Verbal Behav*, *8*, 591-596.
- Pettijohn, K. A., Thompson, A. N., Tamplin, A. K., Krawietz, S. A., & Radvansky, G. A. (2016). Event boundaries and memory improvement. *Cognition*, *148*, 136-144. doi:10.1016/j.cognition.2015.12.013
- Radvansky, G. A., & Copeland, D. E. (2006). Walking through doorways causes forgetting: Situation models and experienced space. *Memory & cognition*, *34*(5), 1150-1156.
- Radvansky, G. A., & Zacks, J. M. (2017). Event Boundaries in Memory and Cognition. *Curr Opin Behav Sci*, *17*, 133-140. doi:10.1016/j.cobeha.2017.08.006
- Rouhani, N., Norman, K. A., Niv, Y., & Bornstein, A. M. (2020). Reward prediction errors create event boundaries in memory. *Cognition*, *203*, 104269. doi:10.1016/j.cognition.2020.104269
- Shin, Y. S., & DuBrow, S. (2020). Structuring memory through inference - based event segmentation. *Topics in Cognitive Science*.
- Sonne, T., Kingo, O. S., & Krojgaard, P. (2017). Bound to remember: Infants show superior memory for objects presented at event boundaries. *Scandinavian journal of psychology*, *58*(2), 107-113. doi:10.1111/sjop.12351
- Swallow, K. M., Zacks, J. M., & Abrams, R. A. (2009). Event boundaries in perception affect memory encoding and updating. *J Exp Psychol Gen*, *138*(2), 236-257. doi:10.1037/a0015631
- Zacks, J. M., Speer, N. K., Swallow, K. M., Braver, T. S., & Reynolds, J. R. (2007). Event perception: a mind-brain perspective. *Psychol Bull*, *133*(2), 273-293. doi:10.1037/0033-2909.133.2.273

REVIEWER COMMENTS

Reviewer #1 (Remarks to the Author):

The revision has thoroughly addressed my concerns regarding the robustness of the results in terms of confounds with list position. That said, to improve clarity, I would recommend motivating why primacy and recency positions were dropped in some analyses and clarifying if/when removal of those items resulted in a reverse bias.

With the clarification of the delta-d metric, it became clear that one strong prediction of the model is that the temporal distance effect (in the absence of boundaries) should get smaller with list position because of the decaying delta-d. That is, it should take much greater distance to achieve an equivalent benefit at the end vs beginning of a boundaryless list. This is also true within events but likely to be harder to detect given their short lengths. Is this prediction supported by the present empirical data? Based on the RT data in Fig S3 and the discussion of scanning models, this does not seem to be the case (i.e., RT does not increase with list position in the no boundary list). Some discussion of this should be included.

The delta-d metric should also negatively predict RT since it is a measure of difficulty of the TOM judgment. It seems that RTs show some of the same patterns that accuracy shows, but not consistently. Of course, RTs may be quite noisy and it is understandable to relegate them to the supplement. That said, there are two analyses I would strongly recommend including in the main text – some measure of speed accuracy tradeoffs and the correlation between delta-d in the model and RT.

I remain concerned about some of the framing with respect to the prior literature:

1. Discussion of previous context models argues that they assume boundary-driven change is “random” but that is not how increased drift rate works. Rather than being random, it is defined by how much new items update the context. (Incidentally, this feature likely also predicts the local primacy effect). Clarification is needed when describing what these previous context models posit.
2. The framing of the finding of both the boundary effect and temporal distance effect as demonstrating that they “can co-exist in the same experiment” still suggests they haven’t been shown together before. The authors should reference the previous work that has found both of these effects in the same experiment.
3. Again, there generally needs to be more care with referencing. As an example, the event horizon model does not make explicit predictions about TOM. That is, interference is an important part of the model but it is not clear how interference between items in an event or interference between events would directly affect TOM (e.g., more interference could lead to more differentiation which could benefit TOM). Moreover, Radvansky & Zacks (2017) appeal to enhanced relational associations within events to explain this effect. Thus, it’s misleading to characterize the event horizon model as making a clear prediction that TOM would be better within events, especially due to interference. Another example – the St Jacques paper (which I had previously recommended for the temporal distance effect) doesn’t do any similarity analysis so how can it show that hippocampal representations are more dissimilar for events that are further apart in time? With fMRI, showing that (at encoding) would be meaningless anyway due to temporal autocorrelation, as I previously noted.

Typo in Fig S7 legend – reference in E to S3A should presumably be S7A.

Reviewer #2 (Remarks to the Author):

I thank the authors for addressing my comments in this revision. I have no further comments.

Reviewer #3 (Remarks to the Author):

I thank the authors for addressing all of my concerns. In so doing, I am even more confident of the significance of the novel empirical results and model. Their modifications have not raised any new concerns, so I am happy to recommend this manuscript for publication.

Response letter to reviewers

Re: Revision of NCOMMS-20-50772-A

Dear Reviewers,

We would like to thank you for all the valuable comments and suggestions. We have answered all your questions and incorporated these suggestions within our manuscript. Our responses are presented in blue after each question. Those changes are highlighted within the manuscript as well.

Again, we thank the reviewers for the insightful comments and close attention to our work.

Sincerely yours,

Yi Pu, Xiangzhen Kong, Charan Ranganath and Lucia Melloni

REVIEWER COMMENTS

Reviewer #1 (Remarks to the Author):

Q1. The revision has thoroughly addressed my concerns regarding the robustness of the results in terms of confounds with list position. That said, to improve clarity, I would recommend motivating why primacy and recency positions were dropped in some analyses and clarifying if/when removal of those items resulted in a reverse bias.

Re: We are delighted to see that we have addressed the reviewer's concerns regarding list position. We have followed the excellent suggestion in the revised manuscript by adding more explanations as to why the first and last positions were dropped (p. 3).

Q2. With the clarification of the delta-d metric, it became clear that one strong prediction of the model is that the temporal distance effect (in the absence of boundaries) should get smaller with list position because of the decaying delta-d. That is, it should take much greater distance to achieve an equivalent benefit at the end vs beginning of a boundaryless list. This is also true within events but likely to be harder to detect given their short lengths. Is this prediction supported by the present empirical data? Based on the RT data in Fig S3 and the discussion of scanning models, this does not seem to be the case (i.e., RT does not increase with list position in the no boundary list). Some discussion of this should be included.

Re: This is an excellent point. We agree with the Reviewer that our model indeed predicts the primacy effect in the long list, i.e., TOM in the early positions should be better than late positions in the long list when controlling for the number of intervening items (Prediction #1). Our model further predicts that the magnitude of the decrease in TOM performance should be larger in the no boundary condition compared to the boundary condition (Prediction #2). This is because in the boundary condition, the resetting process at event boundaries increases delta-d. To test these two predictions, we calculated the average TOM accuracy for early and late [matched] within-event pairs (see Fig. 1 below) for both boundary condition and no boundary condition.

Fig. 1. Schematic diagram showing the early and late positions in the long list for [matched] within-event items.

Then we ran a two (Conditions: boundary condition vs. no boundary condition) by two (Positions: early vs. late) repeated measures ANOVA. Following our model predictions, we expected a Condition by Position interaction. In line our model predictions, we found a significant Condition by Position interaction ($F(1, 25) = 4.601, p=0.0419$). We also found a significant main effect of Condition ($F(1, 25) = 5.291, p=0.0301$) and a significant main effect

of Position ($F(1, 25) = 15.11, p=0.0007$) (see below Fig. 2). That is, in both conditions, there was a decrease in TOM accuracy from early to late positions (early vs. late position in the boundary condition: $t(25)= 2.392, p= 0.0246$, and in the no boundary condition: $t(25)= 5.425, p < 0.0001$, Prediction #1). Critically, consistent with our model Prediction #2, TOM was significantly worse for late position in the no boundary condition compared to the boundary condition (boundary vs. no boundary for late position: $t(25)=3.156, p=0.0041$, while $t(25) = 0.1222, p=0.9037$ for early position). We have added the analysis and the discussion in the Discussion (p. 22) and Supplementary materials Fig. S3 in the revised manuscript.

Fig. 2. Box plot of the accuracy of temporal order memory for early and late positions for the boundary condition and no boundary condition. The whiskers represent the maximum and minimum of the data.

As for the reaction time (RT), we deem them not ideally suited to test the model prediction. As pointed out by the Reviewer in Q3, the RTs in our studies are noisy. This is likely because we instructed subjects to prioritize decision accuracy over RT. We apologize for not being accurate when describing this in Methods section in the previous versions of the manuscript. We have clarified this in the revised manuscript (p. 26). For completion, however, we ran a similar analysis on the RT as for the accuracy. No significant effect was found ($F(1,25)=0.18, p=0.675$ for Condition by Position interaction, $F(1,25)=0.594, p=0.448$ for main effect of Condition, $F(1,25)=2.255, p=0.146$ for main effect of Position).

Q3. The delta-d metric should also negatively predict RT since it is a measure of difficulty of the TOM judgment. It seems that RTs show some of the same patterns that accuracy shows, but not consistently. Of course, RTs may be quite noisy and it is understandable to relegate them to the supplement. That said, there are two analyses I would strongly recommend including in the main text – some measure of speed accuracy tradeoffs and the correlation between delta-d in the model and RT.

Re: Another excellent suggestion. We absolutely agree with the Reviewer that RT should have reflected TOM performance, e.g., difficulty. However, as explained in the response to Q2, since our experimental instruction required participants to prioritize accuracy, the RT measurement may not be as reliable as accuracy to reflect TOM performance. Nevertheless, as suggested, we correlated RT with the memory index (d2-d1) in our model. See Fig. 3 below for results. Though noisy, RT significantly correlates with the memory index in our model across a range of parameter (p and λ) values. i.e., the quicker the RT is, the larger the value of d2-d1 is (Fig. 3C below). And the correlation pattern across parameter values is similar to the correlation pattern between accuracy and the memory index. However, due to the noisiness of RT, the value of R^2 is generally smaller for RT than for accuracy.

Fig. 3. Correlation between model outputs and reaction time (RT) under different parameter values. **A.** Explained variance (R^2) of RT by the model outputs, fitted with a generalized linear model. **B.** Explained variance (R^2) of RT by the model outputs, fitted with a generalized linear mixed model. **C.** Scatter plot of Pearson correlation between model outputs and RT under one parameter value. The black line in images indicates the significance threshold of $p = 0.05$, and the red line indicates the parameter values that yield model outputs which can recover all the effects in the four behavioural experiments.

As for speed accuracy tradeoffs, those might not straightforwardly apply to our data since the task was not setup for participants to respond as accurately and as fast as possible, but instead to prioritize accuracy at the expense of speed (we now have clarified this in Methods p.26). Against that background, we have not included a speed-accuracy tradeoff analysis. Along the same lines,

since RT were not prioritized when the participants performed the task. This might explain why they are noisier than accuracy. Therefore, we opted for placing the analysis on RT in the Supplementary Materials (p. 18 and Fig. S9).

I remain concerned about some of the framing with respect to the prior literature:

Re: We thank the reviewer for carefully reading our manuscript and for all the excellent suggestions. In the revised manuscript, we corrected the inaccuracies pointed out below. We also double checked all the other references cited in the manuscript.

Q4. Discussion of previous context models argues that they assume boundary-driven change is “random” but that is not how increased drift rate works. Rather than being random, it is defined by how much new items update the context. (Incidentally, this feature likely also predicts the local primacy effect). Clarification is needed when describing what these previous context models posit.

Re: Thanks for pointing this out. The Reviewer is right that the increased drift rate (i.e., the magnitude of the change) at event boundaries in previous models (e.g., Horner et al., 2016) depends on how much change occurs in the internal/external environment (Horner et al., 2016). We added this point in the revised manuscript (p.19). Yet, the change (note not the rate of the change) at event boundaries in Horner et al.’ model is random, because a random set of binary elements in the context vector are selected to change (i.e., 0 will change to 1, 1 will change to 0) in the model (p.19). In contrast, the change at event boundaries in our model is not random, because some elements of the context vector from the first context will be recovered at event boundaries. Therefore, the change at event boundaries in our model is towards a reference context (i.e., the first context).

Also, it should be noted that previous model (e.g., Horner et al., 2016) used the contextual similarity of the two probed items as the proxy for TOM. To our knowledge, introducing a sharp change at event boundaries in their model could not capture the local primacy effect (see Fig. 4 below for a model simulation): no clear difference between pairs taking early and late event positions. We added the analysis in p. 23 and Fig. S6 in the revised manuscript.

Our model, in contrast, can simultaneously explain the primacy effect and the local primacy effect on TOM, without further introducing any new parameters. We are indebted to the reviewer’s keen observations and inspiring questions, which have helped us to better present our data and to highlight the novelty and the significance of our manuscript.

Fig. 4. Simulation of the local primacy effect based on Horner et al. (2016)'s model.

Q5. The framing of the finding of both the boundary effect and temporal distance effect as demonstrating that they “can co-exist in the same experiment” still suggests they haven’t been shown together before. The authors should reference the previous work that has found both of these effects in the same experiment.

Re: We added the citation of the previous work as suggested (p. 9).

Q6. Again, there generally needs to be more care with referencing. As an example, the event horizon model does not make explicit predictions about TOM. That is, interference is an important part of the model but it is not clear how interference between items in an event or interference between events would directly affect TOM (e.g., more interference could lead to more differentiation which could benefit TOM). Moreover, Radvansky & Zacks (2017) appeal to enhanced relational associations within events to explain this effect. Thus, it’s misleading to characterize the event horizon model as making a clear prediction that TOM would be better within events, especially due to interference. Another example – the St Jacques paper (which I had previously recommended for the temporal distance effect) doesn’t do any similarity analysis so how can it show that hippocampal representations are more dissimilar for events that are further apart in time? With fMRI, showing that (at encoding) would be meaningless anyway due to temporal autocorrelation, as I previously noted.

Re: We thank the Reviewer for the keen observations of those citations and apologize for those inaccuracies. We agree that interference is a key part in the Event Horizon Model, and does not make an explicit reference to TOM. We have clarified this in the revised manuscript (p.3).

We have removed the St Jacques’s work as one piece of evidence supporting the idea that hippocampal representations are more dissimilar for events that are further apart in time (p.4). We apologize for the mistake.

Q7. Typo in Fig S7 legend – reference in E to S3A should presumably be S7A.

Re: corrected.

Reviewer #2 (Remarks to the Author):

I thank the authors for addressing my comments in this revision. I have no further comments.

Re: Many thanks for the helpful comments during the previous revisions.

Reviewer #3 (Remarks to the Author):

I thank the authors for addressing all of my concerns. In so doing, I am even more confident of the significance of the novel empirical results and model. Their modifications have not raised any new concerns, so I am happy to recommend this manuscript for publication.

Re: We are delighted to see that our previous responses have satisfactorily addressed the concerns; and even more so that they have increased the confidence in the significance and novelty of our results and model.

REVIEWER COMMENTS

Reviewer #1 (Remarks to the Author):

The authors have now fully addressed all of my concerns. The new early/late analyses are particularly compelling additional support for the model. I appreciate their work on the revision and thorough responses.